# *Arabidopsis* formin 2 regulates cell-to-cell trafficking by capping and stabilizing actin filaments at plasmodesmata

Min Diao[1,2,3], Sulin Ren[2], Qiannan Wang[1], Lichao Qian[1,4], Jiangfeng Shen[1], Yule Liu[1,4], Shanjin Huang[1]*

[1]Center for Plant Biology, School of Life Sciences, Tsinghua University, Beijing, China; [2]Institute of Botany, Chinese Academy of Sciences, Beijing, China; [3]University of Chinese Academy of Sciences, Beijing, China; [4]MOE Key Laboratory of Bioinformatics, Tsinghua-Peking Center for Life Sciences, Tsinghua University, Beijing, China

**Abstract** Here, we demonstrate that *Arabidopsis thaliana* Formin 2 (AtFH2) localizes to plasmodesmata (PD) through its transmembrane domain and is required for normal intercellular trafficking. Although loss-of-function *atfh2* mutants have no overt developmental defect, PD's permeability and sensitivity to virus infection are increased in *atfh2* plants. Interestingly, AtFH2 functions in a partially redundant manner with its closest homolog AtFH1, which also contains a PD localization signal. Strikingly, targeting of Class I formins to PD was also confirmed in rice, suggesting that the involvement of Class I formins in regulating actin dynamics at PD may be evolutionarily conserved in plants. In vitro biochemical analysis showed that AtFH2 fails to nucleate actin assembly but caps and stabilizes actin filaments. We also demonstrate that the interaction between AtFH2 and actin filaments is crucial for its function in vivo. These data allow us to propose that AtFH2 regulates PD's permeability by anchoring actin filaments to PD.
DOI: https://doi.org/10.7554/eLife.36316.001

**\*For correspondence:**
sjhuang@tsinghua.edu.cn

**Competing interests:** The authors declare that no competing interests exist.

## Introduction

Plasmodesmata (PD) function as the intercellular channels in plants and are important for the growth and development of plants, as well as during their interaction with the surrounding environment (*Cheval and Faulkner, 2018*; *Lee, 2015*; *Lucas and Lee, 2004*; *Maule, 2008*; *Maule et al., 2011*). The density and size of PD are developmentally controlled (*Xu and Jackson, 2010*), which allows the formation of spatial symplastic domains in order to establish tissue-specific developmental programs. The permeability of PD must be precisely regulated at specific times during plant growth and development. However, the mechanisms that tightly regulate the permeability of PD are largely unknown.

The actin cytoskeleton has been implicated in intercellular communication via PD (*Aaziz et al., 2001*; *Chen et al., 2010*; *Pitzalis and Heinlein, 2017*; *White and Barton, 2011*). In support of this notion, actin and some actin-associated proteins were demonstrated to localize to PD (*Deeks et al., 2012*; *Faulkner et al., 2009*; *Radford and White, 1998*; *Van Gestel et al., 2003*). However, due to technical problems, the existence of filamentous actin in PD remains controversial, as does the nature of its organization. This has been a barrier to our understanding of the function of the actin cytoskeleton in the regulation of cell-to-cell trafficking via PD. In addition, to date, most of the data regarding the function of actin at PD have been obtained from experiments using actin-based pharmacological treatments, which showed that destabilization of actin filaments increases the permeability of PD, whereas stabilization of actin filaments decreases it (*Ding et al., 1996*; *Su et al.,*

**eLife digest** Plant cells communicate with each other via narrow channels embedded across adjacent cell walls. These channels, called plasmodesmata, allow molecules to pass between cells, thereby enabling plants to grow normally and develop tissues and organs. But plasmodesmata also serve as passageways that viruses can exploit to infect more and more cells. Given these pros and cons, plants must regulate how permeable their plasmodesmata are so they can transport necessary materials cell-to-cell while still defending against the spread of infection.

Each cell within plants, animals, and fungi, contains a protein skeleton that helps to stabilize it. A threadlike fiber called actin filament, one of the key components that makes up the cell's skeleton, presumably extends out to the plasmodesmata, which lie across the cell's external wall. Previous research has shown that actin helps regulate cell-to-cell traffic through the plasmodesmata and that drug treatments involving actin disturb normal traffic. But techniques to visualize actin at the plasmodesmata are lacking, and both how plants control their plasmodesmata and actin's involvement remain unclear.

Diao et al. used a confocal microscope, fluorescent tags, and staining procedures in experiments that analyzed how plasmodesmata and actin interact within a small flowering plant called thale cress. These experiments showed that a protein known to regulate actin, called Formin 2, positions itself at the plasmodesmata where it caps off actin threads and anchors them to the channels. Diao et al. also generated thale cress that cannot produce Formin 2. These mutant plants had more permeable plasmodesmata and were more susceptible to a virus.

By stably tethering actin to the plasmodesmata, Formin 2 plays a key part in regulating the permeability of these cell-to-cell channels, with unstable actin threads resulting in more penetrable plasmodesmata. These findings provide further evidence that plants rely on actin to regulate plasmodesmata, and they establish that Formin 2 is involved. Further research will clarify how actin and Formin 2 work together to adjust the structure of plasmodesmata channels.

DOI: https://doi.org/10.7554/eLife.36316.002

2010). Given that these actin drugs non-selectively target the actin cytoskeleton within cells, it is hard to assess whether and to what extent the changes in PD permeability depend on the alteration in PD actin dynamics. In view of this point, specific manipulation of the actin dynamics at PD via genetic means is an ideal approach, although it requires the identification of PD-localized actin or actin-associated proteins that can serve as the target for the genetic manipulation. Nevertheless, previous observations suggest that the presence and dynamics of the actin cytoskeleton are indeed vital for intercellular trafficking via PD. For example, an interesting study showed that *Cucumber mosaic virus* (CMV) Movement Protein (MP) behaves like an actin-binding protein (ABP) that severs and caps actin filaments in vitro and its filament severing activity is required for its function in increasing the size exclusion limit (SEL) of PD (*Su et al., 2010*). The authors therefore argued that actin filaments at PD function as a filter to control the permeability of PD (*Chen et al., 2010*). The study of Chen et al. also suggests that some endogenous ABPs may be involved in regulating actin dynamics at PD and consequently in controlling the permeability of PD.

Formin (formin homology protein) is one type of actin nucleation factor that has been implicated in the generation of linear actin bundles (*Blanchoin and Staiger, 2010*; *Chesarone et al., 2010*; *Kovar, 2006*). The formin proteins are characterized by the presence of formin homology domain1 (FH1) and FH2, which are capable of nucleating actin assembly from actin or actin-profilin complexes (*Blanchoin and Staiger, 2010*; *Chesarone et al., 2010*; *Kovar, 2006*). There are numerous formin genes in plants (*Blanchoin and Staiger, 2010*; *Cvrcková et al., 2004*). For instance, the *Arabidopsis* genome contains 21 formin genes that are divided into two classes: 11 members in Class I, which encode proteins that contain the characteristic transmembrane (TM) domain except *Arabidopsis* formin 7; and 10 members in Class II, which encode proteins without a TM domain but carry an N-terminal phosphatase and tensin-related (PTEN)-like domain (*Blanchoin and Staiger, 2010*; *Deeks et al., 2002*). The Formin proteins have been implicated in numerous actin-based cellular processes in plants, such as polarized pollen tube growth and root hair growth, cell division, cytokinesis and cell morphogenesis, and plant defense (*Cheung et al., 2010*; *Favery et al., 2004*;

*Ingouff et al., 2005*; *Li et al., 2010*; *Yang et al., 2011*; *Ye et al., 2009*; *Zhang et al., 2011*). However, it remains largely unexplored how different formin proteins evolve and adapt to various actin-based cellular functions in plants.

Here we demonstrate that AtFH2, along with several Class I formins, localizes to PD and is involved in the regulation of cell-to-cell trafficking via PD. The PD-localization pattern of AtFH2 is determined by its TM domain. Interestingly, we found that several rice Class I formins also target to PD, suggesting that the involvement of Class I formins in regulating actin dynamics at PD may be evolutionarily conserved in plants. We showed that AtFH2 caps and stabilizes actin filaments in vitro, which allows us to propose that AtFH2 regulates actin dynamics and cell-to-cell trafficking by anchoring and stabilizing actin filaments at PD. Loss of function of *AtFH2* increases PD permeability, which supports the notion that the stability and/or the amount of actin filaments at PD is crucial for their role in regulating intercellular trafficking. Our study thus adds formin as an important component of the actin-mediated machinery that regulates PD permeability.

## Results

### Loss of function of *AtFH2* does not cause overt developmental defects in *Arabidopsis*

To determine the physiological functions of *AtFH2* (At2g43800), two T-DNA insertion mutants of *AtFH2* were obtained and analyzed (*Figure 1—figure supplement 1A*). They were shown to be knockout alleles since the full-length *AtFH2* transcript is absent in both *atfh2* mutants (*Figure 1—figure supplement 1B,C*). However, no overt developmental defects were observed in young seedlings and mature plants of *atfh2* mutants (*Figure 1—figure supplement 1D*). This might be due to functional redundancy among *Arabidopsis* formin genes. However, considering that *Arabidopsis* formin genes exhibit distinct tissue expression patterns and their encoded proteins are quite diverged (*Cvrcková et al., 2004*), they likely perform some distinct and specialized functions that cannot be uncovered by simply examining the morphology of seedlings and adult plants.

### AtFH2 localizes to PD in a manner that depends on its TM domain-containing N-terminus

Determination of the intracellular localization of AtFH2 may provide hints regarding whether AtFH2 performs specialized cellular functions. We therefore generated an eGFP fusion construct of *AtFH2* driven by its own promoter (*AtFH2p:AtFH2-eGFP*) and demonstrated that it is fully functional (see below), which suggests that AtFH2-eGFP will faithfully indicate the intracellular localization of AtFH2. Initial observations revealed that AtFH2-eGFP decorates dot-like structures along the borders of cells (*Figure 1—figure supplement 2A*). Given that AtFH2 contains a TM domain, we speculated that the AtFH2-decorated dot-like structures may be located on the plasma membrane. In partial support of this speculation, the AtFH2-eGFP-decorated structures overlap with the staining of propidium iodide (PI) (*Figure 1—figure supplement 2B*), a nucleic acid binding dye that cannot cross the membrane of living cells and consequently stays outside the membrane, thus revealing the contours of cells. Interestingly, the AtFH2-eGFP-positive dot-like structures are reminiscent of PD. To demonstrate this, we performed callose staining, which has been used previously to detect PD (*Bell and Oparka, 2011*; *Currier, 1957*; *Fitzgibbon et al., 2010*). We found that the AtFH2-eGFP-decorated structures overlap with dot-like structures revealed by callose staining (*Figure 1A*). This result was extended by showing that AtFH2 colocalized with two PD markers, PDLP1-GFP (*Figure 1B*; [*Thomas et al., 2008*]) and YFP-PDCB1 (*Figure 1C*; [*Simpson et al., 2009*]). Thus, the data demonstrate unambiguously that AtFH2 localizes to PD.

Next, we asked how AtFH2 targets to PD. We initially found that the localization pattern of AtFH2 does not depend on the presence of an intact actin cytoskeleton (*Figure 1—figure supplement 2C,D*). This suggests that the PD localization pattern of AtFH2 is not controlled by the C-terminal FH1 and FH2 domains, which together occupy most of the AtFH2 protein and mediate its interaction with the actin cytoskeleton. We speculated that the PD localization pattern of AtFH2 might be solely determined by its TM domain, which is located near the N-terminus. To quickly test this hypothesis, we adopted the strategy of exogenous expression in tobacco leaves. The feasibility of using this strategy was demonstrated by showing that full-length AtFH2 targets to PD in tobacco

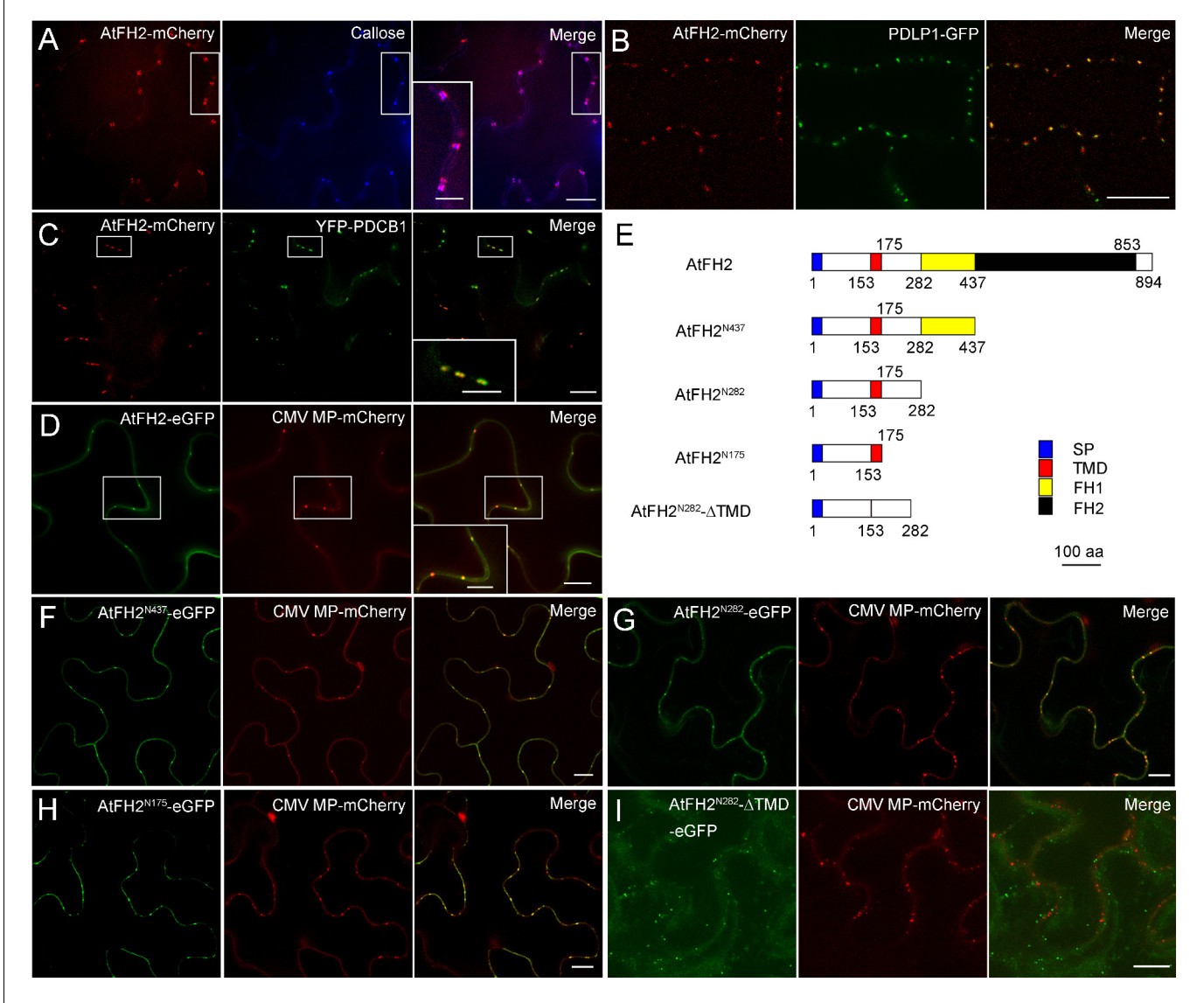

**Figure 1.** AtFH2 localizes to plasmodesmata and its PD localization is determined by the TM-containing N-terminus. (A) Images of AtFH2-mCherry and callose (stained with aniline blue) in *Arabidopsis* epidermal pavement cells and the merged image. (B) Images of AtFH2-mCherry and PDLP1-eGFP in *Arabidopsis* epidermal pavement cells and the merged image. (C) Images of AtFH2-mCherry and YFP-PDCB1 in *Arabidopsis* epidermal pavement cells and the merged image. (D) AtFH2-eGFP, CMV MP-mCherry and the merged image in *N. benthamiana* leaf epidermal cells. (E) Schematic representation of AtFH2 and the truncated proteins used for the intracellular localization analysis. SP, signal peptide; TMD, trans-membrane domain. (F) AtFH2$^{N437}$-eGFP, CMV MP-mCherry and the merged image in epidermal cells of *N. benthamiana* leaves. (G) AtFH2$^{N282}$-eGFP, CMV MP-mCherry and the merged image in epidermal cells of *N. benthamiana* leaves. (H) AtFH2$^{N175}$-eGFP, CMV MP-mCherry and the merged image in epidermal cells of *N. benthamiana* leaves. (I) AtFH2$^{N282}$-ΔTMD-eGFP, CMV MP-mCherry and the merged image in epidermal cells of *N. benthamiana* leaves. Bars = 10 μm in all images, and bars = 5 μm in all inset images.

DOI: https://doi.org/10.7554/eLife.36316.003

The following source data and figure supplements are available for figure 1:

**Figure supplement 1.** No overt developmental defect is detected in seedlings and adult plants of *atfh2* mutants.
DOI: https://doi.org/10.7554/eLife.36316.004

**Figure supplement 1—source data 1.** Quantification of relative expression of AtFH2 plotted in *Figure 1—figure Supplement 1C*.
DOI: https://doi.org/10.7554/eLife.36316.005

**Figure supplement 2.** AtFH2 localizes to cell borders and its localization pattern does not depend on the presence of an intact actin cytoskeleton.
DOI: https://doi.org/10.7554/eLife.36316.006

leaf epidermal cells (*Figure 1D*). We generated several fusion constructs in which eGFP was fused with truncated fragments of AtFH2, starting from the N-terminus (*Figure 1E*). We subsequently demonstrated that eGFP fusion proteins of truncated AtFH2 localize to PD as long as they contain the TM domain (*Figure 1F–H*). The eGFP fusion protein without the TM domain failed to target to PD (*Figure 1I*). Thus, the data suggest that PD-localization of AtFH2 is determined by its TM domain.

## The size exclusion limit (SEL) of PD is increased in *atfh2*

To determine whether cell-to-cell trafficking was altered in *atfh2*, we performed an eGFP diffusion assay as reported previously (*Levy et al., 2007*; *Liarzi and Epel, 2005*). Leaf epidermal cells were transformed by DNA bombardment with an eGFP-expressing construct. After bombardment by plasmids expressing non-mobile HDEL-mCherry along with plasmids expressing eGFP, the HDEL-mCherry signal mainly appeared in one cell (*Figure 2—figure supplement 1A*), which suggests that the eGFP signal outside this cell results from diffusion rather than local expression. The efficiency of eGFP diffusion from bombarded cells into surrounding epidermal layers was assessed by counting the number of cell layers with a diffuse eGFP signal (*Figure 2A*). Our results showed that eGFP diffusion across cells is significantly higher in *atfh2* than in WT (*Figure 2B*; *Figure 2—figure supplement 1B*). The eGFP diffusion phenotype in *atfh2* is rescued by transforming *AtFH2p:AtFH2-eGFP* into *atfh2* (*Figure 2C*), suggesting that increased eGFP diffusion is indeed caused by loss of function of *AtFH2*. Consistent with this notion, the number of eGFP-expressing cells within a diffusion cluster is significantly higher in *atfh2* mutant plants (*Figure 2D*). We found that loss of function of *AtFH2* does not affect cytoplasmic streaming (*Figure 2—figure supplement 2*), which suggests that loss of function of *AtFH2* specifically alters cell-to-cell trafficking. However, we found that overexpression of *AtFH2* does not affect the overall organization of actin filaments or cell-to-cell trafficking in leaf epidermal cells (*Figure 2—figure supplement 3*). Next, we asked whether loss of function of *AtFH2* also affects specific movement of macromolecules across cells by examining the cell-to-cell movement of an eGFP fusion of the *Cucumber mosaic virus* (CMV) protein MP (MP-eGFP). We found that the cell-to-cell movement of MP-eGFP is also increased in *atfh2* (*Figure 2E*). This finding also motivated us to speculate that *atfh2* plants might become more sensitive to virus infection. To test this speculation, we challenged *Arabidopsis* plants with CMV strain Fny (Fny-CMV), which is known to infect *Arabidopsis* (*Du et al., 2014*). We found that *Arabidopsis* leaves became curved and distorted after inoculation with Fny-CMV when compared to the control (*Figure 2—figure supplement 4A,B*). The growth of leaves was inhibited (*Figure 2—figure supplement 4C,D*) and the proportion of symptomatic plants increased in a time-dependent manner (*Figure 2—figure supplement 4E*). Infection with Fny-CMV was confirmed by measurements showing that the amount of CMV *MP* RNA increases in a time-dependent manner (*Figure 2—figure supplement 4F*). We next challenged *Arabidopsis* plants with Fny-CMV and found that the symptoms of infection were more severe in *atfh2* plants than in WT plants (*Figure 2F,G*). This was quantified by the time-dependent increase in the relative proportion of symptomatic plants (*Figure 2H*) and by the relative accumulation of CMV *MP* RNA (*Figure 2I*) in *atfh2* mutants. In addition, the growth of *Arabidopsis* leaves was inhibited more severely in *atfh2* compared to WT plants (*Figure 2J,K*). These data together suggest that the SEL of PD increases in *atfh2* plants.

## The functions of AtFH2 overlap with AtFH1, another PD-localized Class I formin, in regulating cell-to-cell trafficking

The cell-to-cell trafficking phenotype is quite subtle in *atfh2*. We wondered whether this is due to the functional redundancy of AtFH2 with other Class I formins (*Figure 3A*). To determine whether there are additional PD-localized Class I formins in *Arabidopsis*, we examined the localization of the N-terminus of all the other *Arabidopsis* Class I formins except AtFH3 (At4g15200), which is pollen-specific (*Ye et al., 2009*), and AtFH7 (At1g59910), which lacks a TM domain (*Cvrcková et al., 2004*) (*Figure 3A*). We found that AtFH1 (At3g25500), AtFH4 (At1g24150), AtFH8 (At1g70140) and AtFH9 (At5g48360) form obvious dot-like structures along the cell membrane whereas AtFH5 (At5g54650), AtFH6 (At5g67470), AtFH10 (At3g07540) and AtFH11 (At3g05470) are distributed uniformly on the cell membrane (*Figure 3B–I*). To determine whether the AtFH1-, AtFH4-, AtFH8- and AtFH9-decorated dot-like structures are PD, we performed simultaneous callose staining or colocalization with

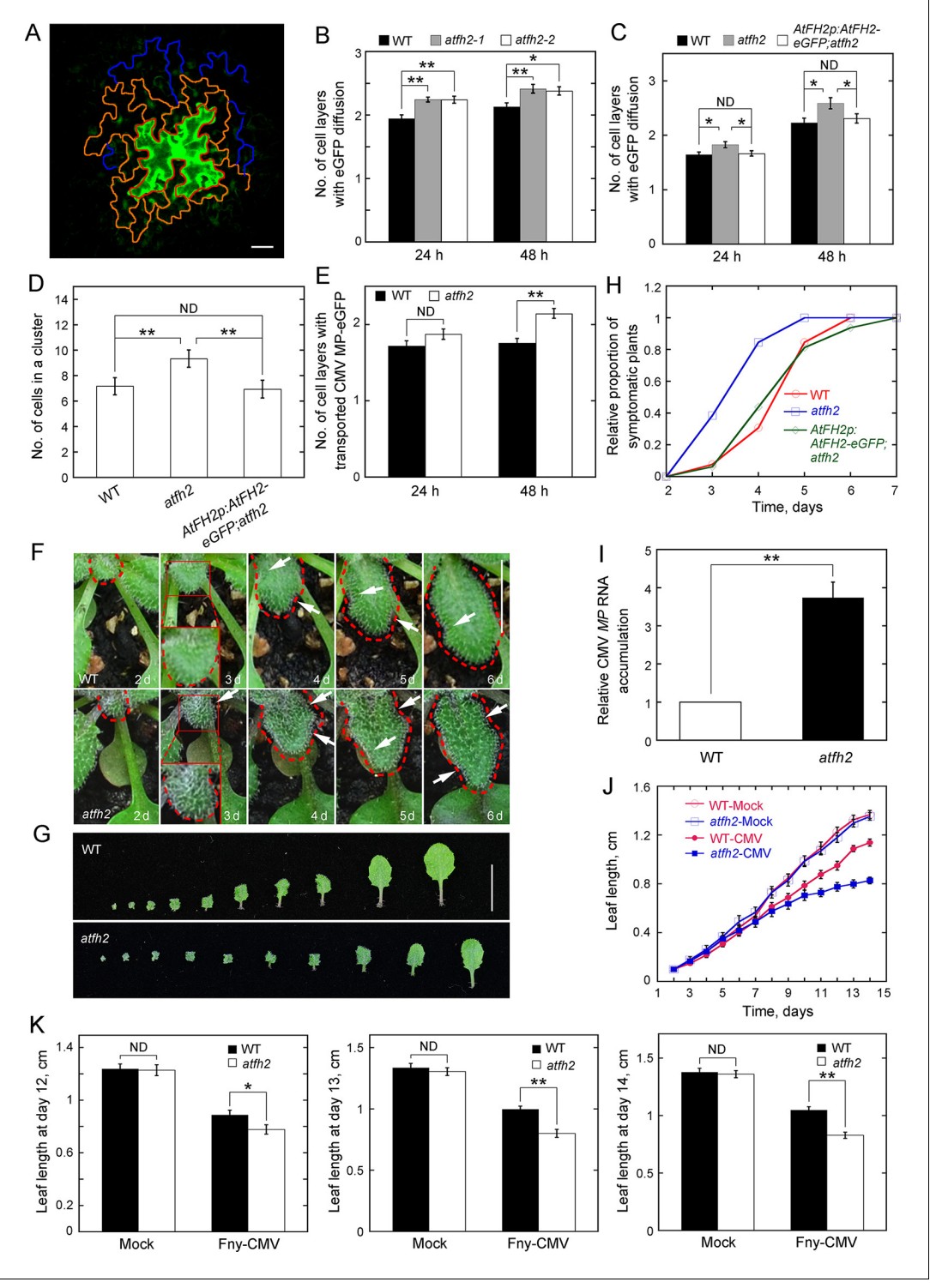

**Figure 2.** Loss of function of *AtFH2* increases SEL of PD. (**A**) Diagram of the eGFP diffusion assay in *Arabidopsis* epidermal pavement cells. The edge of the initial cell expressing eGFP, cells in the first diffusion layer and cells in the second diffusion layer are marked by red, orange and blue lines, respectively. Bar = 10 μm. (**B, C**) Quantification of the number of cell layers showing eGFP diffusion. The number of eGFP diffusion cell layers at 24 hr and 48 hr after the bombardment was counted and plotted. The experiments were repeated at least three times, and more than 30 cells expressing eGFP were counted each time. Values represent mean ± SE. ND, no statistical difference, **p<0.01 and *p<0.05 by Mann-Whitney U test. Given that *atfh2-1* and *atfh2-2* exhibit a similar cell-to-cell trafficking phenotype, we only used *atfh2-1* as the representative *AtFH2* loss-of-function mutant

*Figure 2 continued on next page*

in the following studies. (D) Quantification of the number of cells in an eGFP diffusion cluster. The total number of cells within the cluster was counted and plotted at 24 hr after the bombardment. The experiments were repeated at least three times, and more than 30 GFP-expressing cells were counted each time. Values represent mean ± SE. ND, no statistical difference, **p<0.01 by Mann-Whitney U test. (E) Quantification of the number of cell layers showing cell-to-cell movement of CMV MP-eGFP. The experiments were repeated at least three times, and more than 30 CMV MP-eGFP-expressing cells were counted each time. Values represent mean ± SE. ND, no statistical difference, **p<0.01 by Mann-Whitney U test. (F) Symptoms of Fny-CMV infection in WT and *atfh2* mutant plants. WT *Arabidopsis* leaves start to wrinkle at day 4, whereas *atfh2* mutant leaves start to wrinkle at day 3. Wrinkles are indicated by white arrows. The edges of leaves are marked with dashed red lines. Bar = 0.5 cm. (G) Symptoms in wild type (top) and *atfh2* mutant (bottom) *Arabidopsis* 2 weeks after systemic infection with Fny-CMV. All leaves from *Arabidopsis* plants 14 days after inoculation with Fny-CMV are shown. Bar = 1 cm. (H) Quantification of the Fny-CMV infection rate of WT, *atfh2* mutant and *AtFH2p:AtFH2-eGFP;atfh2* plants. The number of symptomatic plants (with curly leaves) was counted at different days and the relative proportion of symptomatic plants was plotted. Symptoms were evident earlier in *atfh2* mutants than in WT and *AtFH2p:AtFH2-eGFP;atfh2* plants. (I) Determination of the relative level of CMV *MP* transcripts by qRT-PCR in the newly grown small leaves of Fny-CMV-infected WT and *atfh2* mutant plants at day 3. Relative CMV *MP* RNA accumulation represents the relative amount of virus in plants. Values represent mean ± SE, **p<0.01 by Student's *t*-test. (J) Quantification of *Arabidopsis* leaf length 2 weeks after infection with Fny-CMV. Mock (water treatment); CMV (Fny-CMV treatment). The growth of *atfh2* leaves is slower than that of WT leaves after infection with Fny-CMV. Values represent mean ± SE, $n \geq 20$. (K) Quantification of *Arabidopsis* leaf length at days 12, 13 and 14 after inoculation with Fny-CMV. Values represent mean ± SE, $n \geq 20$. **p<0.01 by Student's *t*-test. ND, no statistical difference.
DOI: https://doi.org/10.7554/eLife.36316.007

The following source data and figure supplements are available for figure 2:

**Source data 1.** Quantification of eGFP diffusion plotted in *Figure 2B*.
DOI: https://doi.org/10.7554/eLife.36316.022
**Source data 2.** Quantification of eGFP diffusion plotted in *Figure 2C*.
DOI: https://doi.org/10.7554/eLife.36316.023
**Source data 3.** Quantification of eGFP diffusion plotted in *Figure 2D*.
DOI: https://doi.org/10.7554/eLife.36316.024
**Source data 4.** Quantification of CMV MP transporting plotted in *Figure 2E*.
DOI: https://doi.org/10.7554/eLife.36316.025
**Source data 5.** Quantification of Fny-CMV infection plants plotted in *Figure 2H*.
DOI: https://doi.org/10.7554/eLife.36316.026
**Source data 6.** Quantification of relative CMV MP RNA accumulation plotted in *Figure 2I*.
DOI: https://doi.org/10.7554/eLife.36316.027
**Source data 7.** Quantification of leaf length plotted in *Figure 2J*.
DOI: https://doi.org/10.7554/eLife.36316.028
**Source data 8.** Quantification of leaf length plotted in *Figure 2K*.
DOI: https://doi.org/10.7554/eLife.36316.029
**Figure supplement 1.** Loss of function of *AtFH1* and/or *AtFH2* increases SEL of PD.
DOI: https://doi.org/10.7554/eLife.36316.008
**Figure supplement 1—source data 1.** Quantification of eGFP diffusion plotted in *Figure 2—Figure Supplement 1B*.
DOI: https://doi.org/10.7554/eLife.36316.009
**Figure supplement 1—source data 2.** Quantification of eGFP diffusion plotted in *Figure 2—Figure Supplement 1C*.
DOI: https://doi.org/10.7554/eLife.36316.010
**Figure supplement 2.** No overt defect of cytoplasmic streaming is detected in hypocotyl cells of *atfh2* mutants.
DOI: https://doi.org/10.7554/eLife.36316.011
**Figure supplement 2—source data 1.** Quantification of cytoplasmic streaming plotted in *Figure 2—Figure Supplement 1B*.
DOI: https://doi.org/10.7554/eLife.36316.012
**Figure supplement 3.** Overexpression of *AtFH2* does not affect the overall organization of actin filaments and cell-to-cell trafficking.
DOI: https://doi.org/10.7554/eLife.36316.013

*Figure 2 continued*

**Figure supplement 3—source data 1.** Quantification of relative expression of *AtFH2* plotted in *Figure 2—Figure Supplement 3A*.
DOI: https://doi.org/10.7554/eLife.36316.014

**Figure supplement 3—source data 2.** Quantification of actin filament density plotted in *Figure 2—Figure Supplement 3D*.
DOI: https://doi.org/10.7554/eLife.36316.015

**Figure supplement 3—source data 3.** Quantification of eGFP diffusion plotted in *Figure 2—Figure Supplement 3E*.
DOI: https://doi.org/10.7554/eLife.36316.016

**Figure supplement 4.** The symptoms of Fny-CMV infection in *Arabidopsis* plants.
DOI: https://doi.org/10.7554/eLife.36316.017

**Figure supplement 4—source data 1.** Quantification of leaf length plotted in *Figure 2—figure supplement 4C*.
DOI: https://doi.org/10.7554/eLife.36316.018

**Figure supplement 4—source data 2.** Quantification of leaf length plotted in *Figure 2—figure supplement 4D*.
DOI: https://doi.org/10.7554/eLife.36316.019

**Figure supplement 4—source data 3.** Quantification of Fny-CMV infection plants plotted in *Figure 2—figure supplement 4E*.
DOI: https://doi.org/10.7554/eLife.36316.020

**Figure supplement 4—source data 4.** Quantification of relative CMV MP RNA accumulationin *Figure 2—figure supplement 4F*.
DOI: https://doi.org/10.7554/eLife.36316.021

AtFH2 (*Figure 3J–O*) and confirmed that AtFH1 and AtFH9 localize to PD (*Figure 3J,K,N and O*). To determine whether the localization of Class I formins to PD is conserved in plants, we examined the intracellular localization of Class I formins in monocotyledon rice. Among 11 Class I rice formins (*Figure 3—figure supplement 1A*; *Cvrcková et al., 2004*), we found that OsFH8 (Os03g0204100), OsFH11 (Os07g0545500), OsFH15 (Os09g0517600) and OsFH16 (Os02g0739100) are able to localize to PD (*Figure 3—figure supplement 1B–E*). Given that AtFH1 and AtFH2 belong to the same subclass (*Figure 3A*), we generated *atfh1 atfh2* double mutants (*Figure 4—figure supplement 1*) to examine whether they redundantly regulate cell-to-cell trafficking via PD. We did not detect overt developmental phenotypes in *atfh1 atfh2* mutants compared to WT, *atfh1* and *atfh2* plants (*Figure 4—figure supplement 1*). However, we found that *atfh1 atfh2* mutants have stronger cell-to-cell trafficking phenotypes than *atfh1* and *atfh2* single mutants (*Figure 4*; *Figure 2—figure supplement 1C*). This suggests that AtFH1 and AtFH2 indeed act redundantly in regulating cell-to-cell trafficking via PD. Given that the accumulation of callose was shown to be involved in the regulation of PD permeability (*Cui and Lee, 2016*; *Han et al., 2014*; *Simpson et al., 2009*), we wondered whether loss of function of *AtFH1* and/or *AtFH2* might alter the actin dynamics at PD and consequently affect the accumulation of callose at PD. We therefore performed callose staining with aniline blue and found that the amount of callose accumulated at PD in *atfh1*, *atfh2* and *atfh1 atfh2* mutants was not overtly different to that in WT plants (*Figure 4—figure supplement 2*). This suggests that the alteration in cell-to-cell trafficking in *atfh1*, *atfh2* and *atfh1 atfh2* mutants is not due to an alteration in the accumulation of callose at PD.

## AtFH2 fails to nucleate actin assembly from actin or actin bound to profilin in vitro

We next generated several recombinant AtFH2 proteins, with increasingly large N-terminal deletions, to determine their effect on actin dynamics in vitro (*Figure 5A,B*). We initially tested whether they are able to nucleate actin assembly, which is a characteristic feature of formins (*Chesarone et al., 2010*; *Goode and Eck, 2007*). The *bona fide* actin nucleator AtFH1 (*Michelot et al., 2005*), which efficiently enhances actin assembly, was used as a positive control (*Figure 5C*). Unexpectedly, AtFH2 recombinant proteins failed to nucleate actin assembly; instead, they slightly inhibited actin assembly (*Figure 5C*). This was confirmed by directly visualizing the effect of AtFH2 recombinant proteins on actin assembly by total internal reflection fluorescence microscopy (TIRFM) (*Figure 5D,E*). In addition, we found that recombinant AtFH2 proteins failed to

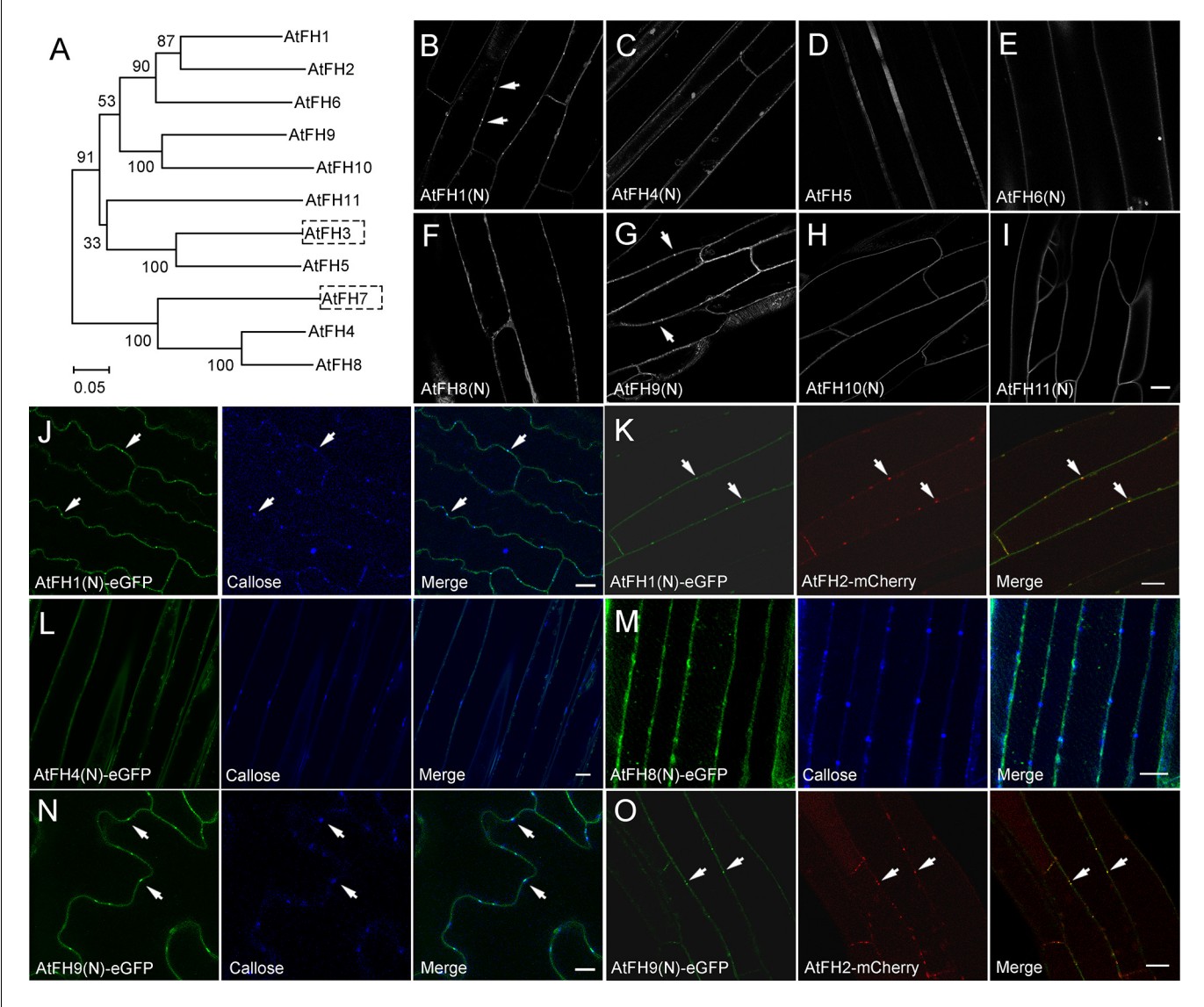

**Figure 3.** Determination of the intracellular localization of *Arabidopsis* Class I formins. (A) Phylogenetic analysis of Class I *Arabidopsis* formins. The phylogenetic tree was made by MEGA 4.0 based on neighbor joining analysis. Numbers on branches represent the bootstrap values. (B–I) Examination of the intracellular localization pattern of Class I *Arabidopsis* formins by expressing their eGFP fusion constructs in *Arabidopsis* hypocotyl cells. (B) AtFH1(N)-eGFP; (C) AtFH4(N)-eGFP; (D) AtFH5-eGFP; (E) AtFH6(N)-eGFP; (F) AtFH8(N)-eGFP; (G) AtFH9(N)-eGFP; (H) AtFH10(N)-eGFP; (I) AtFH11(N)-eGFP. White arrows indicate punctate structures. (J) AtFH1(N)-eGFP, callose, and the merged image in *Arabidopsis* expressing *35S:AtFH1(N)-eGFP*. (K) AtFH1(N)-eGFP, AtFH2-mCherry, and the merged image in *Arabidopsis* plants expressing *35S:AtFH1(N)-eGFP* and *35S:AtFH2-mCherry*. (L) AtFH4(N)-eGFP, callose, and the merged image in *Arabidopsis* expressing *35S:AtFH4(N)-eGFP*. (M) AtFH8(N)-eGFP, callose, and the merged image in *Arabidopsis* expressing *35S:AtFH8(N)-eGFP*. (N) AtFH9(N)-eGFP, callose, and the merged image in *Arabidopsis* expressing *35S:AtFH9(N)-eGFP*. (O) AtFH9(N)-eGFP, AtFH2-mCherry, and the merged image in *Arabidopsis* plants expressing *35S:AtFH9(N)-eGFP* and *AtFH2pro:AtFH2-mCherry*. White arrows in (J), (K), (N) and (O) indicate PD. Bars = 10 μm.

DOI: https://doi.org/10.7554/eLife.36316.030

The following figure supplement is available for figure 3:

**Figure supplement 1.** Several Class I rice formins localize to plasmodesmata.

DOI: https://doi.org/10.7554/eLife.36316.031

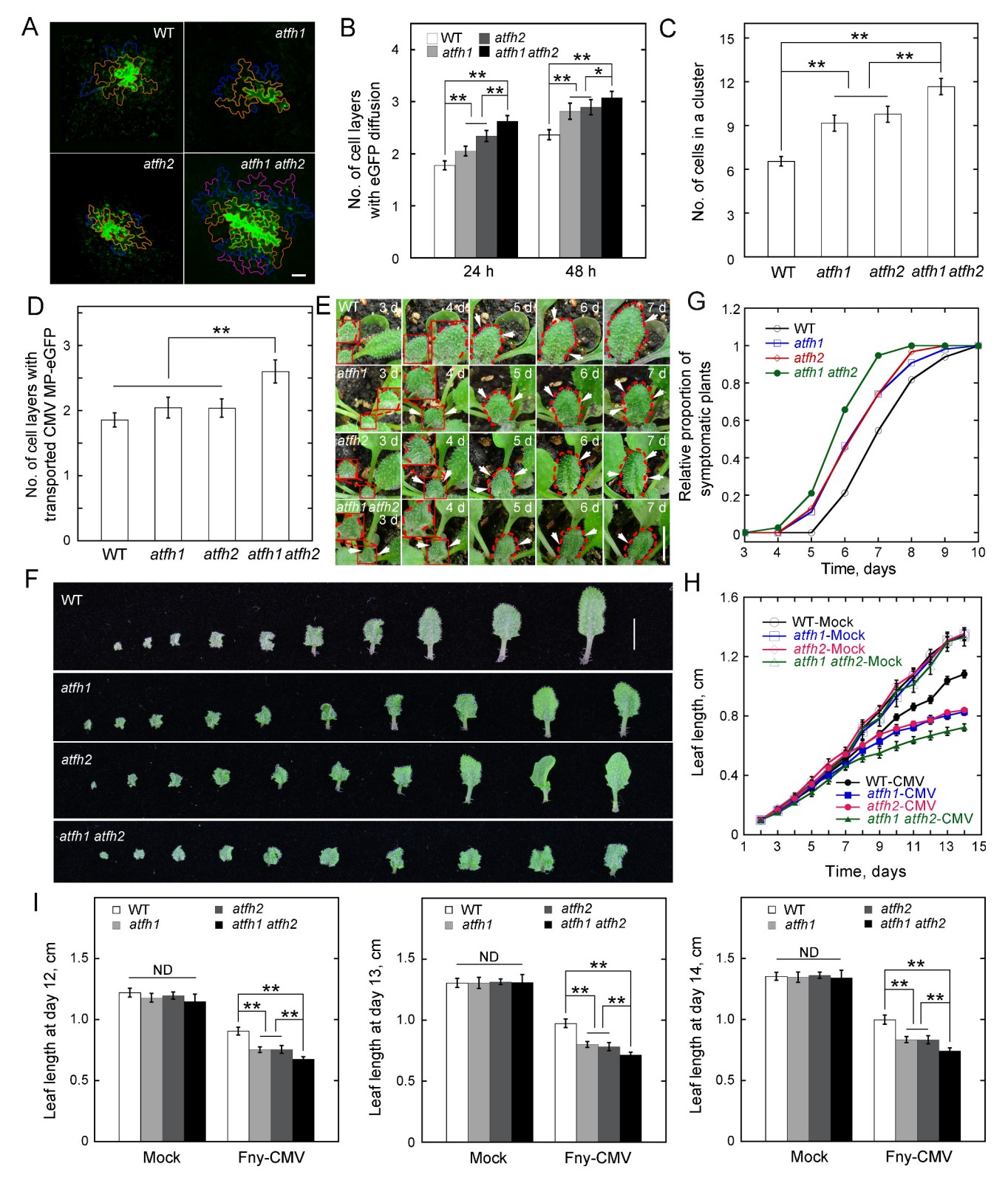

**Figure 4.** AtFH1 and AtFH2 redundantly regulate PD permeability. (**A**) Images of eGFP diffusion in leaf epidermal pavement cells of WT, *atfh1*, *atfh2* and *atfh1 atfh2* plants. The edge of the cell that initially expressed eGFP is marked by a red line. The edge of cells within the first, second and third diffusion layers are marked by orange, blue, and pink lines, respectively. Bar = 10 μm. (**B**) Quantification of the number of eGFP diffusion layers in leaves of WT, *atfh1*, *atfh2* and *atfh1 atfh2* plants. The number of cell layers showing eGFP diffusion at 24 hr and 48 hr was measured and plotted. Values

*Figure 4 continued on next page*

*Figure 4 continued*

represent mean ± SE. *p<0.05, and **p<0.01 by Mann-Whitney U test. More than 30 cells were counted each time and the experiments were repeated at least three times. (C) Quantification of the number of cells in an eGFP diffusion cluster in WT, *atfh1*, *atfh2* and *atfh1 atfh2* mutant plants. The number of cells was determined at 24 hr after the bombardment. Values represent mean ± SE. **p<0.01 by Mann-Whitney U test. More than 30 cells were counted each time and the experiments were repeated at least three times. (D) Quantification of the number of cell layers showing cell-to-cell movement of CMV MP-eGFP in leaves of WT, *atfh1*, *atfh2* and *atfh1 atfh2* plants. The number of diffusion cell layers at 48 hr was measured and plotted. Values represent mean ± SE. **p<0.01, by Mann-Whitney U test. More than 30 cells were counted and the experiments were repeated at least three times. (E) Symptom of Fny-CMV infection in WT, *atfh1*, *atfh2* and *atfh1 atfh2* mutant *Arabidopsis* plants. Small leaves started to become wrinkled (indicated by white arrows) between days 3 to 7. The edges of leaves are marked with dashed red lines. Bar = 1 cm. (F) Images of *Arabidopsis* leaves from WT, *atfh1*, *atfh2* and *atfh1 atfh2* plants 14 days after inoculation with Fny-CMV. Bar = 1 cm. (G) Quantification of the relative proportion of symptomatic plants in WT, *atfh1*, *atfh2* and *atfh1 atfh2* plants after infection with Fny-CMV. The number of symptomatic plants (with curly leaves) was counted at different days and the relative proportion of symptomatic plants was plotted. (H) Quantification of *Arabidopsis* leaf length after Fny-CMV infection. The length of leaves was defined according to the method shown in *Figure 2—figure supplement 4C*. The average length of *Arabidopsis* leaves was traced for 2 weeks. (I) Quantification of *Arabidopsis* leaf length at days 12, 13 and 14 after inoculation. Values represent mean ± SE, $n \geq 20$. **p<0.01 by Student's *t*-test. ND, no statistical difference.
DOI: https://doi.org/10.7554/eLife.36316.032

The following source data and figure supplements are available for figure 4:

**Source data 1.** Quantification of eGFP diffusion plotted in *Figure 4B*.
DOI: https://doi.org/10.7554/eLife.36316.042
**Source data 2.** Quantification of eGFP diffusion plotted in *Figure 4C*.
DOI: https://doi.org/10.7554/eLife.36316.043
**Source data 3.** Quantification of CMV MP transporting plotted in *Figure 4D*.
DOI: https://doi.org/10.7554/eLife.36316.044
**Source data 4.** Quantification of Fny-CMV infection plants plotted in *Figure 4G*.
DOI: https://doi.org/10.7554/eLife.36316.045
**Source data 5.** Quantification of leaf length plotted in *Figure 4H*.
DOI: https://doi.org/10.7554/eLife.36316.046
**Source data 6.** Quantification of leaf length plotted in *Figure 4I*.
DOI: https://doi.org/10.7554/eLife.36316.047
**Figure supplement 1.** Characterization of *atfh1* and *atfh1 atfh2* mutants.
DOI: https://doi.org/10.7554/eLife.36316.033
**Figure supplement 1—source data 1.** Quantification of relative expression of *AtFH1* plotted in *Figure 4—figure supplement 1C*.
DOI: https://doi.org/10.7554/eLife.36316.034
**Figure supplement 1—source data 2.** Quantification of circularity plotted in *Figure 4—figure supplement 1D*.
DOI: https://doi.org/10.7554/eLife.36316.035
**Figure supplement 1—source data 3.** Quantification of eGFP diffusion plotted in *Figure 4—figure supplement 1E*.
DOI: https://doi.org/10.7554/eLife.36316.036
**Figure supplement 1—source data 4.** Quantification of relative expression of *AtFH1* plotted in *Figure 4—figure supplement 1F*.
DOI: https://doi.org/10.7554/eLife.36316.037
**Figure supplement 1—source data 5.** Quantification of relative expression of *AtFH2* plotted in *Figure 4—figure supplement 1G*.
DOI: https://doi.org/10.7554/eLife.36316.038
**Figure supplement 2.** Callose accumulation is not affected in *atfh1*, *atfh2* and *atfh1 atfh2* mutants.
DOI: https://doi.org/10.7554/eLife.36316.039
**Figure supplement 2—source data 1.** Quantification of the size of aniline blue-stained foci in *Figure 4—figure supplement 2B*.
DOI: https://doi.org/10.7554/eLife.36316.040
**Figure supplement 2—source data 2.** Quantification of the gray values of aniline blue-stained foci in *Figure 4—figure supplement 2C*.
DOI: https://doi.org/10.7554/eLife.36316.041

nucleate actin assembly from profilin-actin complexes (*Figure 5F*). Thus, our results showed that AtFH2 fails to nucleate actin assembly from actin or actin bound to profilin in vitro.

## AtFH2 caps the barbed end of actin filaments and stabilizes them from depolymerization

To determine whether AtFH2 has the characteristic barbed end capping activity of formins, we performed the seeded actin elongation assay. We found that recombinant AtFH2-ΔN, AtFH2-FH1FH2 and AtFH2-FH2 prevented the addition of profilin-actin complexes in a dose-dependent manner

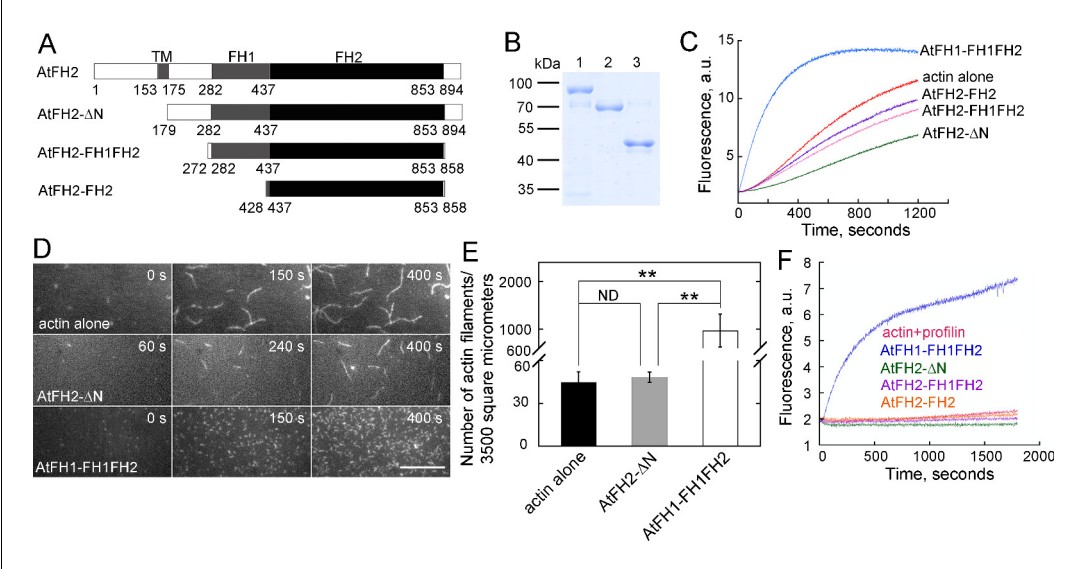

**Figure 5.** AtFH2 fails to nucleate actin assembly from actin or actin bound to profilin, and slightly inhibits spontaneous actin polymerization in vitro. (**A**) Schematic representation of the domain organization of AtFH2 and the fragments used for the generation of recombinant truncated AtFH2 proteins. (**B**) Purified recombinant truncated AtFH2 proteins. The recombinant AtFH2 proteins were resolved on 10% SDS-PAGE and stained with Coomassie blue. Lane 1, AtFH2-ΔN; lane 2, AtFH2-FH1FH2; lane 3, AtFH2-FH2. (**C**) The effect of AtFH2 on spontaneous actin assembly from G-actin alone. Actin (10% pyrene-labeled, 3 μM) was incubated with recombinant formin proteins (1 μM) for 5 min at room temperature before the addition of 10 × KMEI to initiate actin polymerization, and actin polymerization was traced by monitoring the changes in pyrene fluorescence. (**D**) Time-lapse images of actin filaments in the absence or presence of formin proteins. [Actin], 1.5 μM (33.3% Oregon Green-labeled); [AtFH2-ΔN], 400 nM; [AtFH1-FH1FH2], 100 nM. Bar = 10 μm. (**E**) Quantification of the number of actin filaments per microscope field. Values represent mean ± SE, *n* = 3. **p<0.01 by Student's *t*-test. (**F**) AtFH2 fails to utilize profilin-actin complexes. Actin (10% pyrene-labeled, 3 μM) plus human profilin I (3 μM) were incubated with recombinant formin proteins (1 μM) for 5 min at room temperature before the addition of 10 × KMEI to initiate actin polymerization, and actin polymerization was traced by monitoring the changes in pyrene fluorescence. AtFH1-FH1FH2 (*Michelot et al., 2005*) was used as the control in (**C**) and (**F**).
DOI: https://doi.org/10.7554/eLife.36316.048

The following source data is available for figure 5:

**Source data 1.** Quantification of number of actin filaments plotted in *Figure 5E*.
DOI: https://doi.org/10.7554/eLife.36316.049

(*Figure 6A,B*), suggesting that AtFH2 does indeed have barbed end capping activity. From three independent experiments, the average dissociation constant ($K_d$) (mean ± SE, *n* = 3) was determined to be 80.3 ± 12.9 nM, 163.3 ± 22.2 nM and 354.6 ± 49.0 nM for AtFH2-ΔN, AtFH2-FH1FH2 and AtFH2-FH2, respectively. The barbed end capping activity of AtFH2 was also confirmed by showing that AtFH2 blocks the annealing of actin filaments (*Figure 6C,D*) and inhibits their elongation (*Figure 6E,F*). Similar to the features reported for other barbed end capping proteins, we found that AtFH2 protects actin filaments from dilution-mediated actin depolymerization in a dose-dependent manner (*Figure 6G*). Thus, our results demonstrated that AtFH2 caps the barbed end of actin filaments and stabilizes them in vitro.

## Interaction between AtFH2 and actin filaments is crucial for the function of AtFH2 in regulating cell-to-cell trafficking in *Arabidopsis*

To directly examine whether the involvement of AtFH2 in the regulation of cell-to-cell trafficking is via its interaction with the actin cytoskeleton, we genetically manipulated AtFH2 in order to alter its interaction with actin filaments. It was reported that substitution of Ile1431 with alanine (A) or of Lys1601 with aspartic acid (D) within the FH2 domain of the yeast formin Bni1p completely abolished the actin binding activity of its FH2 domain since it fails to form dimers (*Xu et al., 2004*). We therefore mutated the corresponding residues in AtFH2 in order to inhibit its dimer formation and disrupt its interaction with actin filaments. We found that two residues, Ile519 and Lys672, in AtFH2 correspond to the conserved Ile1431 and Lys1601 in Bni1p (*Figure 7A*). We initially replaced Ile519 with

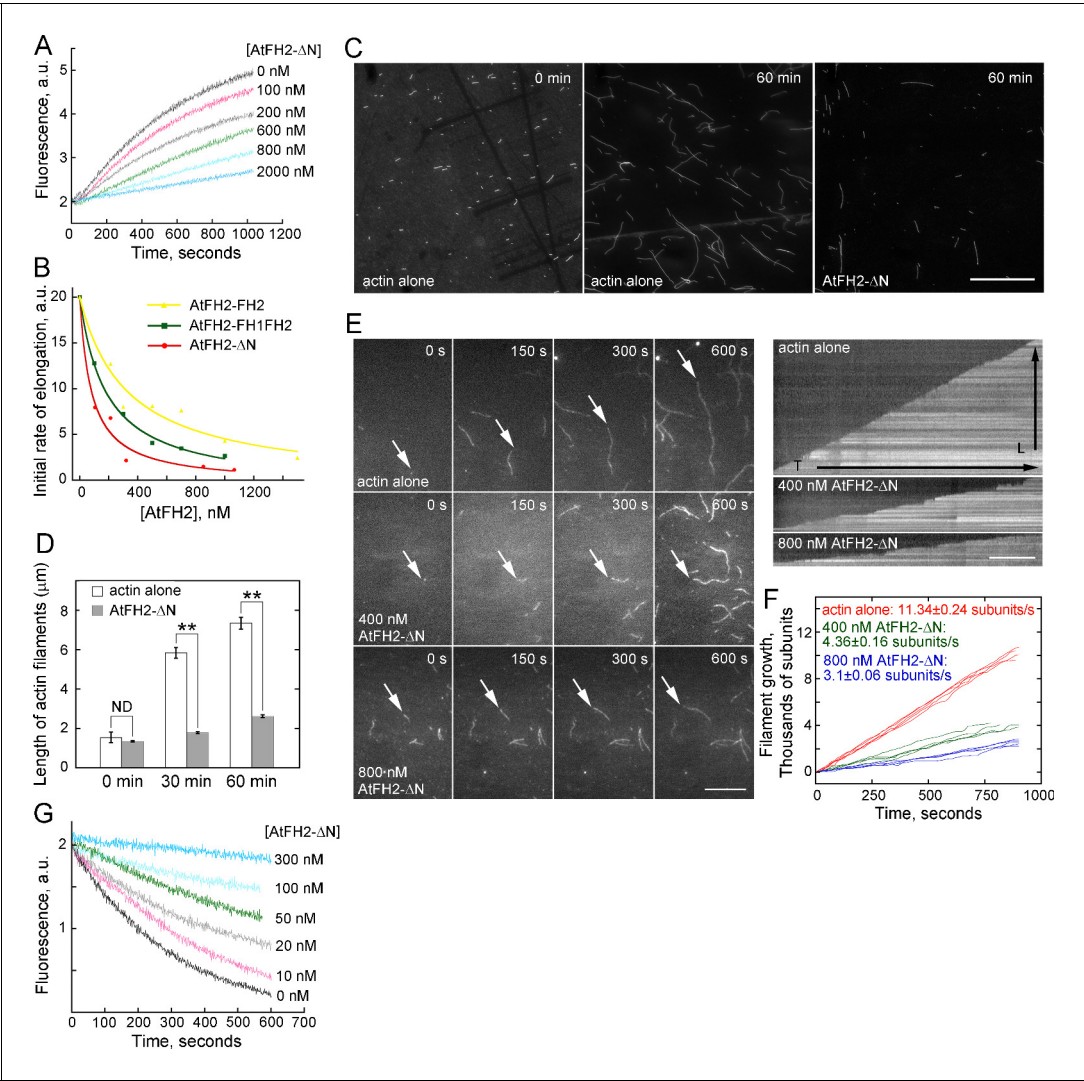

**Figure 6.** AtFH2 caps the barbed end of actin filaments and stabilizes them from dilution-mediated actin depolymerization. (A) AtFH2-ΔN inhibits actin barbed end elongation in a dose-dependent manner. Pre-formed F-actin seeds (0.8 μM) were incubated with different concentrations of AtFH2-ΔN for 5 min at room temperature, and 1 μM G-actin (10% pyrene-labeled) saturated with 4 μM human profilin I was subsequently added to initiate actin elongation at the barbed end. (B) Determination of the dissociation constant ($K_d$) for the interaction between AtFH2 recombinant proteins and the barbed end of actin filaments. The initial rates of elongation were plotted versus the concentration of recombinant formin proteins (AtFH2-ΔN, AtFH2-FH1FH2 and AtFH2-FH2). The $K_d$ values were calculated to be 80 nM, 183 nM and 329 nM for AtFH2-ΔN, AtFH2-FH1FH2 and AtFH2-FH2, respectively. (C) Micrographs of actin filaments before and after annealing. Preassembled actin filaments (4 μM) labeled with equimolar rhodamine-phalloidin were sheared with needles in the absence or presence of AtFH2-ΔN (2 μM). Bar = 10 μm. (D) Quantification of the length of actin filaments before and after annealing. The average length of actin filaments at 0 min, 30 min and 60 min was plotted; values represent mean ± SE, $n \geq 100$. **p<0.01 by Student's *t*-test. ND, no statistical difference. (E) Time-lapse images of actin filaments in the absence or presence of recombinant formin proteins (left panel). White arrows indicate the elongating end of actin filaments. The right panels are the kymograph analyses of growing actin filaments shown in the left panels. [Actin], 1.5 μM (33.3% Oregon Green-labeled); [AtFH2-ΔN], 400 nM; [AtFH1-FH1FH2], 400 nM. Bars = 10 μm. (F) Quantification of the elongation rates of actin filaments in the absence or presence of recombinant formin proteins. The average elongation rates (mean ± SE, $n \geq 20$; subunits/s) of actin filaments are 11.34 ± 0.24, 4.36 ± 0.16 and 3.1 ± 0.06 for actin alone, actin +400 nM AtFH2-ΔN and actin +800 nM AtFH2-ΔN, respectively. (G) Plot of dilution-induced actin depolymerization curves. Various concentrations of AtFH2-ΔN were incubated with preformed actin filaments (5 μM, 50% pyrene-labelled) for 5 min at room temperature before being diluted 25-fold into Buffer G.

DOI: https://doi.org/10.7554/eLife.36316.050

The following source data is available for figure 6:

**Source data 1.** Quantification of of the dissociation constant (Kd) of AtFH2 plotted in *Figure 6B*.
DOI: https://doi.org/10.7554/eLife.36316.051
**Source data 2.** Quantification of the length of actin filaments plotted in *Figure 6D*.

*Figure 6 continued on next page*

*Figure 6 continued*

DOI: https://doi.org/10.7554/eLife.36316.052
**Source data 3.** Quantification of the elongation rates of actin filaments plotted in *Figure 6F*.
DOI: https://doi.org/10.7554/eLife.36316.053

A and Lys672 with D in AtFH2-FH2 and generated the recombinant AtFH2-FH2$^{I519A,K672D}$ protein (*Figure 7B*; named as AtFH2-FH2M hereafter). We found that, unlike AtFH2-FH2, AtFH2-FH2M failed to inhibit actin nucleation (*Figure 7C*) and actin filament elongation (*Figure 7D*). In contrast to AtFH2-FH2, which is mainly in the dimeric form, AtFH2-FH2M is mainly in the monomeric form (*Figure 7E*). These data together suggest that AtFH2-FH2M fails to dimerize and interact with actin filaments. To determine how the mutation affects the role of AtFH2 in regulating PD-mediated cell-to-cell trafficking, we initially found that AtFH2M-eGFP still localizes to PD (*Figure 7F*). In contrast to AtFH2-eGFP, we found that AtFH2M failed to rescue the cell-to-cell trafficking phenotype in *atfh2* mutants assayed with eGFP diffusion (*Figure 7G*) and virus infection (*Figure 7H*) experiments. Thus, these data together suggest that the regulatory function of AtFH2 on PD permeability depends on its interaction with the actin cytoskeleton.

## Discussion

Here, we demonstrate that the Class I formin AtFH2 localizes to PD through its TM domain and is involved in the regulation of cell-to-cell trafficking via PD. Given that AtFH2 is a simple actin filament barbed end capper that stabilizes actin filaments in vitro (*Figures 5* and *6*), we propose that AtFH2 acts as an anchor that tethers actin filaments to the membrane at PD and stabilizes the filaments through its barbed end capping activity. In the absence of AtFH2, the amount of actin filaments is assumed to be reduced at PD. In this regard, our findings that loss of function of *AtFH2* increases the SEL of PD (*Figures 2* and *4*) are consistent with previous results, obtained from actin-based pharmacological treatments, showing that destabilization of actin filaments increases the SEL of PD (*Ding et al., 1996*; *Su et al., 2010*). These data together allow us to conclude that the stability and/ or the amount of actin filaments is crucial for their role in regulating the permeability of PD. Our study thus significantly enhances our understanding of actin-mediated regulation of intercellular trafficking via PD in plants.

Decades of studies have confirmed the presence of the actin cytoskeletal system at PD and demonstrated its involvement in the regulation of PD-mediated intercellular trafficking (*Ding et al., 1996*; *Fernandez-Calvino et al., 2011*; *Su et al., 2010*), but the underlying molecular and cellular mechanisms remain largely unknown. Studies in this area have progressed slowly for at least two reasons. First, we still lack approaches to specifically visualize actin at PD since they are small and buried deeply in the cell wall. Consequently, the existence of filamentous forms of actin at PD remains controversial. Second, we still lack approaches to specifically perturb actin dynamics at PD. Identification of the native components of the actin cytoskeletal system that specifically localize to PD may provide targets for the genetic manipulation of the actin cytoskeleton in this region, which will enhance our understanding of how actin at PD regulates cell-to-cell trafficking.

Identification of the localization of AtFH2 to PD and demonstration of its involvement in PD permeability open a window for us to dissect the status, organization and functions of actin at PD. Several lines of evidence show that loss of function of *AtFH2* increases the SEL of PD (*Figures 2* and *4*; *Figure 2—figure supplement 1B,C*). Although we cannot directly visualize the organization of actin at PD in WT or *atfh2* plants, we speculate that the amount of F-actin is reduced at PD in *atfh2* plants since AtFH2 caps and stabilizes actin filaments even though it lacks the characteristic actin nucleation activity (*Figures 5* and *6*). Our data allow us to propose that PD-localized AtFH2 recruits actin filaments to PD by capping the barbed end of actin filaments and stabilizing them, which consequently regulates cell-to-cell trafficking. Given that AtFH2 is a simple actin barbed end capper that stabilizes actin filaments in vitro (*Figures 5* and *6*) and loss of function of *AtFH2* increases the permeability of PD (*Figures 2* and *4*), it is likely that there are fewer actin filaments at PD in *atfh2* mutants, so that intercellular trafficking is upregulated. Our results, along with previous data from actin-based pharmacological treatments showing that destabilization of actin filaments increases SEL

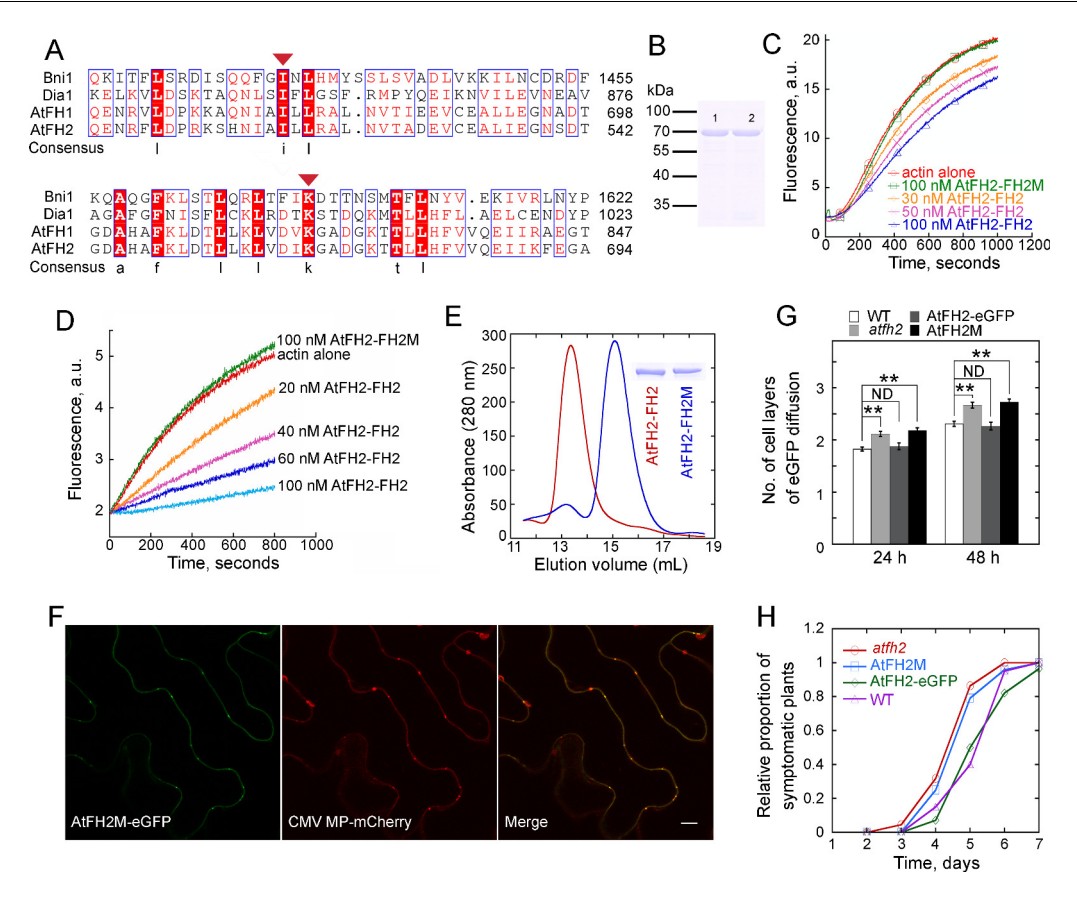

**Figure 7.** Interaction of AtFH2 with actin filaments is required for its function in regulating the permeability of PD. (**A**) Protein sequence alignment of the FH2 domains in AtFH2 and three other well-characterized formins. Ile519 (I) and Lys672 (K) in AtFH2 (red triangles) correspond to Ile1431 and Lys1601 in Bni1p. These residues were previously shown to be important for the dimer formation and actin nucleation activity of Bni1p (*Xu et al., 2004*). *S. cerevisiae* Bni1p (Bni1), P41832; human diaphanous protein 1 (Dia1), O60610; AtFH1, Q9SE97. (**B**) SDS-PAGE analysis of recombinant AtFH2-FH2 and AtFH2-FH2M. Lane 1, GST-AtFH2-FH2; lane 2, GST-AtFH2-FH2M. (**C**) The effect of AtFH2-FH2 and AtFH2-FH2M on spontaneous actin assembly. The conditions for actin assembly are described in *Figure 5C*. (**D**) The effect of AtFH2-FH2 and AtFH2-FH2M on seeded actin elongation. The conditions for actin assembly are described in *Figure 6A*. (**E**) Elution volume of AtFH2-FH2 (red line) and AtFH2-FH2M (blue line) by size exclusion chromatography. To exclude the potential interference of GST in dimer formation, His-tag fusion proteins were generated. The inset shows SDS-PAGE analysis of 6His-AtFH2-FH2 and 6His-AtFH2-FH2M. The estimated molecular weight is about 100 kDa for AtFH2-FH2 and 50 kDa for AtFH2-FH2M. (**F**) Subcellular localization of AtFH2M-eGFP and CMV MP-mCherry in epidermal pavement cells of *N. benthamiana* leaves. Bar = 10 μm. (**G**) Quantification of eGFP diffusion cell layers in *Arabidopsis* leaf epidermal cells. AtFH2M represents *AtFH2p:AtFH2M;atfh2*, and AtFH2-eGFP represents *AtFH2p:AtFH2-eGFP; atfh2*. The number of diffusion cell layers at 24 hr and 48 hr was plotted. Error bars represent SE, *n* > 30, **p<0.01 by Mann-Whitney U test. ND, no statistical difference. The experiment was repeated at least three times. (**H**) Quantification of Fny-CMV infection rates in *Arabidopsis* plants. The number of symptomatic plants (with curly leaves) was counted at different days and the relative proportion of symptomatic plants was plotted.

DOI: https://doi.org/10.7554/eLife.36316.054

The following source data is available for figure 7:

**Source data 1.** Quantification of eGFP diffusion plotted in *Figure 7G*.
DOI: https://doi.org/10.7554/eLife.36316.055
**Source data 2.** Quantification of Fny-CMV infection plants plotted in *Figure 7H*.
DOI: https://doi.org/10.7554/eLife.36316.056

of PD (*Ding et al., 1996*; *Su et al., 2010*), suggest that actin filaments at PD may act as the physical barrier to regulate its permeability (*Chen et al., 2010*). To gain further insights into the function of actin filaments in the regulation of the permeability of PD, it will be necessary in the future to determine where AtFH2-tethered actin filaments localize along the transport path of PD. The reconstructed three-dimensional ultrastructure of post-cytokinesis plasmodesmata using electron tomography showed that the gap between the plasma membrane (PM) and the ER (called the

cytoplasmic sleeve) within plasmodesmata pores is less than 10 nm (*Nicolas et al., 2017*). This suggests that there is unlikely to be enough space to fit more actin filaments within the cytoplasmic sleeve of PD. The authors also discovered a second PD morphotype (type I) that lacks a visible cytoplasmic sleeve but is capable of non-targeted movement of macromolecules (*Nicolas et al., 2017*), which urges us to rethink the relationship between the size of the cytoplasmic sleeve and the permeability. Nonetheless, these findings suggest that the contact between ER and PM is very dynamic. In this regard, AtFH2-tethered actin filaments might be involved in the regulation of the dynamic connection between ER and PM, and may consequently modulate the permeability of PD, although the underlying mechanism remains largely unknown at the moment. Certainly, it might also be possible that AtFH2-tethered actin filaments localize at the neck region of PD to regulate cell-to-cell transportation. Again, it remains to be established how AtFH2-tethered actin filaments at the neck of PD act as a barrier in cell-to-cell trafficking.

It is intriguing that loss of function of both *AtFH1* and *AtFH2* does not have an overt effect on *Arabidopsis* growth and development (*Figure 4—figure supplement 1H,I*), even though it significantly increases the permeability of PD (*Figure 4A–D*). The biological significance of formin-mediated regulation of PD permeability is supported by the finding that loss of function of *AtFH1* and/or *AtFH2* increases the sensitivity of *Arabidopsis* plants to infection by Fny-CMV (*Figure 4E–I*). This suggests that the processes involved in normal plant development may be less sensitive to changes in formin-mediated regulation of PD permeability than the processes involved in coping with biotic stress. Given that AtFH2 is a simple actin barbed end capper (*Figures 5* and *6*), it is still an outstanding question how the barbed end capping activity of AtFH2 is tightly regulated in response to developmental and environmental cues in order to fine-tune the permeability of PD.

Surprisingly, and unlike previously characterized formins (*Blanchoin and Staiger, 2010*; *Chesarone et al., 2010*; *Goode and Eck, 2007*; *Kovar, 2006*), AtFH2 lacks actin nucleation activity in the presence or absence of profilin but retains the actin filament barbed end capping activity in vitro (*Figures 5* and *6*). The crystal structure of the FH2 domain derived from yeast Bni1p showed that it forms a stable yet flexible dimer (*Xu et al., 2004*). The later co-crystal of Bni1p FH2 in complex with actin demonstrates the flexibility of the FH2 dimer, which permits the addition of actin monomers onto the barbed end of actin filaments (*Otomo et al., 2005*). Based on the structural and biochemical data, it was proposed that the FH2 dimer exists in a dynamic equilibrium between a 'closed' and an 'open' state (*Kovar, 2006*). It is possible that FH2 dimers in AtFH2 might be less flexible, and are able to cap the barbed end of actin filaments but cannot facilitate the further addition of actin or actin-profilin complexes. In the future, careful examination of the molecular details underlying the unique activity of AtFH2 may provide insights into the mechanism of action of formin in regulating actin nucleation and elongation. Nonetheless, based on the unique biochemical properties of AtFH2, we propose that it serves well as an anchor for tethering actin filaments at the membrane of PD by capping the barbed end of actin filaments. Since AtFH2 presumably only interacts with filamentous actin through its barbed end capping activity (*Figures 5* and *6*), and the interaction of AtFH2 with actin filaments was demonstrated to be crucial for its in vivo functions (*Figure 7G,H*), this study to some extent provides another piece of evidence supporting the presence of filamentous actin at PD.

Interestingly, we found that AtFH1, the closest homolog of AtFH2, is also able to target to PD (*Figure 3*) and functions redundantly with AtFH2 in the regulation of intercellular trafficking (*Figure 4*). Biochemically, their functional redundancy is likely due to the fact that both have capping activity (*Figure 6*; [*Michelot et al., 2005*]). Considering that AtFH1 is a non-processive actin nucleation factor (*Michelot et al., 2006*), AtFH1 and AtFH2 might coordinately regulate actin dynamics at PD, with AtFH1 promoting actin polymerization to generate more actin filaments around PD, and AtFH2 subsequently taking over the binding to the barbed end of actin filaments after the dissociation of AtFH1. We cannot completely rule out the possibility that AtFH2 might have actin nucleation activity when bound to other proteins and/or after post-translational modification in vivo. More work needs to be undertaken to examine these possibilities in the future. Nonetheless, our findings provide insights into the functional specification of members of the formin protein family and the functional adaptation of the actin cytoskeletal system in plants in general.

Our results showed that the localization of AtFH2 is determined by its TM domain (*Figure 1*), which suggests that the TM domain of AtFH2 has evolved to allow it to adapt to regulate actin dynamics at PD. Given that the membrane of PD has a unique phospholipid composition

(*Grison et al., 2015*), the TM domains of AtFH1 and AtFH2 may bind preferentially to certain phospholipid(s) when compared to other Class I formins in *Arabidopsis*. Strikingly, we showed that several rice Class I formins localize to PD (*Figure 3—figure supplement 1B–E*), suggesting that the involvement of Class I formins in regulating actin dynamics at PD might be evolutionarily conserved in plants.

In summary, we found that AtFH2 localizes to PD and functions redundantly with AtFH1 to regulate cell-to-cell trafficking in *Arabidopsis*. Based on the in vitro biochemical data that AtFH2 caps and stabilizes actin filaments, we propose that AtFH2 regulates cell-to-cell trafficking by tethering actin filaments to the membrane at PD and stabilizing them through its barbed end capping activity. Our study suggests that the stability and/or the amount of actin filaments at PD is crucial for the permeability of PD.

# Materials and methods

**Key resources table**

| Reagent type (species) or resource | Designation | Source or reference | Identifiers | Additional information |
|---|---|---|---|---|
| Gene (*Arabidopsis thaliana*) | *AtFH1* | PMID: 14671023 | TAIR: At3g25500 | |
| Gene (*Arabidopsis thaliana*) | *AtFH2* | this paper | TAIR: At2g43800 | |
| Gene (*Arabidopsis thaliana*) | *AtFH4* | PMID: 16313636 | TAIR: At1g24150 | |
| Gene (*Arabidopsis thaliana*) | *AtFH5* | PMID: 15765105 | TAIR: At5g54650 | |
| Gene (*Arabidopsis thaliana*) | *AtFH6* | PMID: 15319477 | TAIR: At5g67470 | |
| Gene (*Arabidopsis thaliana*) | *AtFH8* | PMID: 16313636 | TAIR: At1g70140 | |
| Gene (*Arabidopsis thaliana*) | *AtFH9* | NA | TAIR: At5g48360 | |
| Gene (*Arabidopsis thaliana*) | *AtFH10* | NA | TAIR: At3g07540 | |
| Gene (*Arabidopsis thaliana*) | *AtFH11* | NA | TAIR: At3g05470 | |
| Gene (*Oryza sativa*) | *OsFH8* | PMID: 15256004 | RAP-DB: Os03g0204100 | |
| Gene (*Oryza sativa*) | *OsFH11* | PMID: 15256004 | RAP-DB: Os07g0545500 | |
| Gene (*Oryza sativa*) | *OsFH15* | PMID: 15256004 | RAP-DB: Os09g0517600 | |
| Gene (*Oryza sativa*) | *OsFH16* | PMID: 15256004 | RAP-DB: Os02g0739100 | |
| Genetic reagent (*Arabidopsis thaliana*) | *atfh2-1* | this paper | GK_066D02 | |
| Genetic reagent (*Arabidopsis thaliana*) | *atfh2-2* | this paper | GK_396H03 | |
| Genetic reagent (*Arabidopsis thaliana*) | *atfh1-1* | PMID: 23202131 | Salk_032981 | |
| Genetic reagent (*Arabidopsis thaliana*) | *atfh1-3* | this paper | Salk_143939 | |
| Strain, strain background (*Escherichia coli*) | DH5α | PMID: 6345791 | | |

*Continued on next page*

*Continued*

| Reagent type (species) or resource | Designation | Source or reference | Identifiers | Additional information |
|---|---|---|---|---|
| Strain, strain background (*Agrobacterium tumefaciens*) | GV3101 | other | | widely distributed |
| Strain, strain background (*Escherichia coli*) | Tuner (DE3) (pLysS) | other | | Novagen, Schwalbach, Germany |
| Transfected construct | pdGN (vector) | PMID: 16126836 | | |
| Transfected construct | *pdGN-35S:HDEL-mCherry* | this paper | | vector-promoter: protein construct |
| Transfected construct | pCambia1301 (vector) | other | GenBank: AF234297.1 | binary vector |
| Transfected construct | pBI101 (vector) | other | GenBank: U12639.1 | binary vector |
| Biological sample (*Arabidopsis thaliana*) | *PDLP1pro:PDLP1-GFP* | PMID: 18215111 | | |
| Biological sample (*Arabidopsis thaliana*) | *PDCB1pro:YFP-PDCB1* | PMID: 19223515 | | |
| Biological sample (*Arabidopsis thaliana*) | *AFH2p:AFH2-eGFP; atfh2* | this paper | | promoter:eGFP fusion protein; *atfh2-1* mutant background |
| Biological sample (*Arabidopsis thaliana*) | *AFH2p:AFH2-mCherry; atfh2* | this paper | | promoter:eGFP fusion protein; *atfh2-1* mutant background |
| Biological sample (*Arabidopsis thaliana*) | *35S:AFH2-mCherry; Col-0* | this paper | | promoter:eGFP fusion protein; Col-0 background |
| Biological sample (*Arabidopsis thaliana*) | *35S:AtFH1(N)-eGFP; Col-0* | this paper | | promoter:eGFP fusion protein; Col-0 background |
| Biological sample (*Arabidopsis thaliana*) | *35S:AtFH4(N)-eGFP; Col-0* | this paper | | promoter:eGFP fusion protein; Col-0 background |
| Biological sample (*Arabidopsis thaliana*) | *35S:AtFH5-eGFP; Col-0* | this paper | | promoter:eGFP fusion protein; Col-0 background |
| Biological sample (*Arabidopsis thaliana*) | *35S:AtFH6(N)-eGFP; Col-0* | this paper | | promoter:eGFP fusion protein; Col-0 background |
| Biological sample (*Arabidopsis thaliana*) | *35S:AtFH8(N)-eGFP; Col-0* | this paper | | promoter:eGFP fusion protein; Col-0 background |
| Biological sample (*Arabidopsis thaliana*) | *35S:AtFH9(N)-eGFP; Col-0* | this paper | | promoter:eGFP fusion protein; Col-0 background |
| Biological sample (*Arabidopsis thaliana*) | *35S:AtFH10(N)-eGFP; Col-0* | this paper | | promoter:eGFP fusion protein; Col-0 background |
| Biological sample (*Arabidopsis thaliana*) | *35S:AtFH11(N)-eGFP; Col-0* | this paper | | promoter:eGFP fusion protein; Col-0 background |
| Biological sample (*Arabidopsis thaliana*) | *35S:GFP-ABD2; Col-0* | PMID: 15557099 | | promoter:eGFP fusion protein; Col-0 background |

*Continued on next page*

*Continued*

| Reagent type (species) or resource | Designation | Source or reference | Identifiers | Additional information |
|---|---|---|---|---|
| Biological sample (*Arabidopsis thaliana*) | *AFH2p:AFH2M; atfh2* | this paper | | promoter:recombinant protein; *atfh2-1* background |
| Recombinant DNA reagent | *pGD-CMV MP-mCherry* | PMID: 20435906 | | |
| Recombinant DNA reagent | *pBI101-35S:AtFH2$^{N175}$-eGFP-NOS* | this paper | | vector-promoter:protein construct |
| Recombinant DNA reagent | *pBI101-35S:AtFH2$^{N282}$-eGFP-NOS* | this paper | | vector-promoter:protein construct |
| Recombinant DNA reagent | *pBI101-35S:AtFH2$^{N437}$-eGFP-NOS* | this paper | | vector-promoter:protein construct |
| Recombinant DNA reagent | *pBI101-35S:AtFH2$^{N282}$-TMD-eGFP-NOS* | this paper | | vector-promoter:protein construct |
| Recombinant DNA reagent | *pBI101-35S:OsFH8(N)-eGFP-NOS* | this paper | | vector-promoter:protein construct |
| Recombinant DNA reagent | *pBI101-35S:OsFH11(N)-eGFP-NOS* | this paper | | vector-promoter:protein construct |
| Recombinant DNA reagent | *pBI101-35S:OsFH15(N)-eGFP-NOS* | this paper | | vector-promoter:protein construct |
| Recombinant DNA reagent | *pBI101-35S:OsFH16(N)-eGFP-NOS* | this paper | | vector-promoter:protein construct |
| Recombinant DNA reagent | *pCambia1301-35S:AtFH2M-eGFP-NOS* | this paper | | vector-promoter:protein construct |
| Recombinant protein | AtFH1-FH1FH2 | PMID: 15994911 | Q9SE97 | |
| Recombinant protein | 6His-AtFH2-ΔN | this paper | | |
| Recombinant protein | 6His-AtFH2-FH1FH2 | this paper | | |
| Recombinant protein | 6His-AtFH2-FH2 | this paper | | |
| Recombinant protein | 6His-AtFH2-FH2M | this paper | | |
| Recombinant protein | GST-AtFH2-FH2 | this paper | | |
| Recombinant protein | GST-AtFH2-FH2M | this paper | | |
| Chemical compound, drug | latrunculin B | Calbiochem | 428020 | |
| Software, algorithm | ImageJ | https://imagej.nih.gov/ij/ | | version 1.51 |
| Software, algorithm | IBM SPSS Statistics | other | | version 25 |

## Plant materials and growth conditions

T-DNA insertion lines GK_066D02, GK_396H03, Salk_032981 and Salk_143939 were designated as *atfh2-1*, *atfh2-2*, *atfh1-1* and *atfh1-3*, respectively. Information about the mutant *atfh1-1* has been presented previously (*Rosero et al., 2013*). Double mutant *atfh1 atfh2* was generated by crossing

*atfh2-1* with *atfh1-3*, and *atfh1-3 atfh2-1* is named as *atfh1 atfh2* throughout the manuscript. *Arabidopsis* Columbia-0 (Col-0) ecotype was used as wild-type (WT) and *Arabidopsis* plants were grown in soil or media under a 16-h-light/8-h-dark photoperiod at 22°C. *Nicotiana benthamiana* plants were grown in pots placed in growth-rooms at 25°C under a 16-h-light/8-h-dark cycle.

## Identification of T-DNA insertion mutants of *AtFH1* and *AtFH2*

The genotyping of *atfh2-1* and *atfh2-2* was performed with primer pairs *atfh2-1* LP/*atfh2-1* RP and *atfh2-2* LP/*atfh2-2* RP (*Supplementary File 1*) in combination with left border primer GK_LB (*Supplementary File 1*), and the genotyping of *atfh1-1* and *atfh1-3* was performed with primer pairs *atfh1-1* LP/*atfh1-1* RP and *atfh1-3* LP/*atfh1-3* RP (*Supplementary File 1*) in combination with left border primer Salk_LB (*Supplementary File 1*), respectively. RT-PCR was performed to determine the transcript levels of *AtFH1* and *AtFH2* in their loss-of-function mutants. Total RNA was isolated from leaves of 4-week-old plants using Trizol reagent (Invitrogen) and reverse transcribed using MMLV reverse transcriptase (Promega) according to the manufacturer's instructions. To perform semi-quantitative RT-PCR analysis, two primer pairs $AtFH2_{CDSFOR}$/$AtFH2_{CDSREV}$ and $AtFH1_{CDSFOR}$/$AtFH1_{CDSREV}$ (*Supplementary File 1*) were used to amplify full-length *AtFH2* and *AtFH1* from WT and their loss-of-function mutants. To perform quantitative RT-PCR analysis, two primer pairs Q1/Q2 and Q3/Q4 (*Supplementary File 1*) were used to determine the transcript levels of *AtFH2* and *AtFH1*, respectively. *eIF4A* was used as an internal control and was amplified with primers $eIF4A_{FOR}$/$eIF4A_{REV}$ (*Supplementary File 1*). The $2^{-\Delta\Delta Ct}$ method was used to quantify the qRT-PCR results. 2 × RealStar Power SYBR Mixture (GeneStar) was used for the amplification.

## Plasmid construction

AtFH2-eGFP fusion constructs, driven either by the *AtFH2* promoter or the *35S* promoter were generated to determine the subcellular localization of AtFH2. The promoter of *AtFH2* was amplified with $AtFH2_{PROFOR}$ and $AtFH2_{PROREV}$ primers (*Supplementary File 1*) using *Arabidopsis* genomic DNA as the template, and the coding sequence (CDS) of *AtFH2* was amplified from WT *Arabidopsis* cDNA using $AtFH2_{CDSFOR}$ and $AtFH2_{CDSREV}$ primers (*Supplementary File 1*). The promoter and CDS were cloned into *pEASY-Blunt* vector (TransGen). The CDS of *AtFH2* was subsequently moved into *pCAMBIA1301-35S:eGFP-NOS* or *pCAMBIA1301-35S:mCherry-NOS* to generate *pCAMBIA1301-35S:AtFH2-eGFP-NOS* or *pCAMBIA1301-35S:AtFH2-mCherry-NOS* plasmids, or moved into *pCAMBIA1301-eGFP-NOS* or *pCAMBIA1301-mCherry-NOS* along with the promoter of *AtFH2* to generate *pCAMBIA1301-AtFH2p:AtFH2-eGFP-NOS* or *pCAMBIA1301-AtFH2p:AtFH2-mCherry-NOS* plasmids. Substitution of isoleucine-519 with alanine (I519A) and lysine-672 with aspartic acid (K672D) was achieved by changing AT to GC and AAA to GAT by PCR with primers containing the corresponding nucleotide changes (M1/M2 and M3/M4; see *Supplementary File 1*) using *pEASY-Blunt-AtFH2* plasmid as the template. The resulting *pEASY-Blunt-AtFH2M* plasmid was digested with *Sal*I/*Bam*HI and the resulting *AtFH2M* fragment was moved into *pCAMBIA1301-35S:eGFP-NOS* or *pCAMBIA1301-NOS* along with the promoter of *AtFH2* to generate *pCAMBIA1301-35S:AtFH2M-eGFP-NOS* and *pCAMBIA1301-AtFH2p:AtFH2M-NOS*. Those plasmids were introduced into *Agrobacterium tumefaciens* strain GV3101 and transformed into either WT or *atfh2* plants via the floral dip method (*Clough and Bent, 1998*). The T3 homozygous plants were used for subsequent analyses.

To determine whether PD localization of AtFH2 is determined by its N-terminus, three N-terminal fragments, designated as $AtFH2^{N437}$, $AtFH2^{N282}$ and $AtFH2^{N175}$, were amplified with primer pairs, $AtFH2_{CDSFOR}$/$AtFH2^{N437}_{REV}$, $AtFH2_{CDSFOR}$/$AtFH2^{N282}_{REV}$ and $AtFH2_{CDSFOR}$/$AtFH2^{N175}_{REV}$ (see *Supplementary File 1*), respectively, using *AtFH2* cDNA as the template. They were subsequently moved into *pBI101-35S:eGFP-NOS* to generate *pBI101-35S:AtFH2^{N437}-eGFP-NOS*, *pBI101-35S:AtFH2^{N282}-eGFP-NOS* and *pBI101-35S:AtFH2^{N175}-eGFP-NOS* plasmids. To determine whether the TM domain of AtFH2 is necessary for its PD localization, the N282 fragment without the TM domain (named as $AtFH2^{N282}$-ΔTMD) was amplified with primer pairs $AtFH2^{N282}$-$ΔTMD_{FOR}$/$AtFH2^{N282}$-$ΔTMD_{REV}$ and $AtFH2_{CDSFOR}$/$AtFH2^{N282}_{REV}$ using *pBI101-35S:AtFH2^{N282}-eGFP-NOS* as the template and the amplified fragment was subsequently moved into *pBI101-35S:eGFP-NOS* to generate *pBI101-35S:AtFH2^{N282}-ΔTMD-eGFP-NOS*. To determine whether other Class I *Arabidopsis* formins localize to PD, the N-terminus of *AtFH1*, *AtFH4*, *AtFH6*, *AtFH8*, *AtFH9*, *AtFH10* and *AtFH11* was

amplified from WT *Arabidopsis* cDNA using primer pairs $AtFH1(N)_{FOR}/AtFH1(N)_{REV}$, $AtFH4(N)_{FOR}/AtFH4(N)_{REV}$, $AtFH6(N)_{FOR}/AtFH6(N)_{REV}$, $AtFH8(N)_{FOR}/AtFH8(N)_{REV}$, $AtFH9(N)_{FOR}/AtFH9(N)_{REV}$, $AtFH10(N)_{FOR}/AtFH10(N)_{REV}$ and $AtFH11(N)_{FOR}/AtFH11(N)_{REV}$ (see *Supplementary File 1*), respectively. Error-free PCR fragments digested with *Sal*I/*Bam*HI were moved into *pBI101-35S:eGFP-NOS* restricted with *Sal*I/*Bam*HI to generate the corresponding plasmids. To determine whether some rice Class I formins localize to PD, the N-terminus of Class I OsFH8, OsFH11, OsFH15 and OsFH16 was amplified from *Oryza sativa* Japonica cDNA using primer pairs $OsFH8(N)_{FOR}/OsFH8(N)_{REV}$, $OsFH11(N)_{FOR}/OsFH11(N)_{REV}$, $OsFH15(N)_{FOR}/OsFH15(N)_{REV}$ and $OsFH16(N)_{FOR}/OsFH16(N)_{REV}$ (*Supplementary File 1*), respectively, and the error-free PCR fragments were subsequently moved into *pBI101-35S:eGFP-NOS* restricted with *Xba*I/*Bam*HI. The resulting eGFP fusion constructs were transiently expressed in *Nicotiana benthamiana* leaves via *Agrobacterium*-mediated transformation (see below). To indicate the primary cell targeted by particle bombardment, the ER marker HDEL (*Batoko et al., 2000*) was amplified with primers $HDEL-mCherry_{FOR}$ and $HDEL-mCherry_{REV}$ (*Supplementary File 1*) using the plasmid *1301-Lat52:HDEL-mCherry-NOS* as the template and subsequently moved into a *pdGN* vector to replace eGFP (*Lee et al., 2005*).

To generate recombinant AtFH2 proteins, three fragments, AtFH2-ΔN, AtFH2-FH1FH2 and AtFH2-FH2, were amplified with primer pairs $AtFH2\Delta N_{FOR}/AtFH2\Delta N_{REV}$, $AtFH2-FH1FH2_{FOR}/AtFH2-FH1FH2_{REV}$ and $AtFH2-FH2_{FOR}/AtFH2-FH2_{REV}$ (*Supplementary File 1*), respectively, using *AtFH2* cDNA as the template. The amplified fragments were moved into pET28a or pET23a to generate prokaryotic expression plasmids, *pET28a-AtFH2-ΔN*, *pET28a-AtFH2-FH1FH2* and *pET23a-AtFH2-FH2*, respectively. For substitution of isoleucine-519 with alanine (I519A) and lysine-672 with aspartic acid (K672D) in AtFH2-FH2, *AtFH2-FH2* was amplified with primer pairs M5/M6 and M7/M8 (*Supplementary File 1*) using *pEASY-Blunt-AtFH2M* as the template. The amplified fragment, named as *AtFH2-FH2M*, was moved into *pGEX-KG* or *pET23b* to generate *pGEX-KG-AtFH2-FH2M* or *pET23b-AtFH2-FH2M* plasmid. They were transformed into *E. coli* Tuner (DE3) pLysS strain.

## Transient expression in *Nicotiana benthamiana*

To determine the colocalization of CMV MP with AtFH2, AtFH2M and rice formins, *Agrobacterium*-mediated transient expression was performed in *N. benthamiana*, which was essentially according to the previously published method (*Voinnet et al., 2003*). *Agrobacterium tumefaciens* stain GV3101 transformed with *pGD-CMV MP-mCherry* (*Su et al., 2010*), *pCambia1301-35S:AtFH2-eGFP-NOS*, *pCambia1301-35S:AtFH2M-eGFP-NOS*, *pBI101-35S:OsFH8(N)-eGFP-NOS*, *pBI101-35S:OsFH11(N)-eGFP-NOS*, *pBI101-35S:OsFH15(N)-eGFP-NOS*, *pBI101-35S:OsFH16(N)-eGFP-NOS* and GV3101 strain carrying p19 were grown at 28°C in Luria Broth culture medium containing 50 μg/mL kanamycin, 50 μg/mL rifampicin, 10 mM MES and 40 μM acetosyringone for 16 hr. Cells were collected by centrifugation and resuspended in 10 mM $MgCl_2$ and 200 μM acetosyringone and the cell suspension was adjusted to $OD_{600}$ = 1.0. Cells were left at room temperature for 3 hr. Cultures were mixed with GV3101 transformed with *pGD-CMV MP-mCherry* or p19 and subsequently infiltrated into 3-week-old *N. benthamiana* leaves. *N. benthamiana* leaf epidermal cells were visualized and the images were captured by laser scanning confocal microscopy after 48–60 hr.

## Microscopy observation of the localization of AtFH2

The localization of fluorescently tagged proteins was assessed using a laser scanning confocal microscope. eGFP or YFP was excited by a 488 nm argon laser and emission was captured in the range of 505–545 nm. mCherry or propidium iodide (PI) was excited by a 543 nm HeNe laser and emission was captured in the range of 590–625 nm. *Arabidopsis* seedlings were cultured on 1/2 Murashige and Skoog culture medium for 3 to 5 days before being observed under the microscope. For PI staining, *Arabidopsis* seedlings were incubated with 50 μg/mL PI (Sigma-Aldrich, P4864) diluted with the solution containing 9% glucose and 5% glycerol for 5 min at room temperature as described previously (*Wang and Huang, 2014*). To demonstrate whether AtFH2 localizes to PD, PD was revealed by staining with aniline blue (see below) or colocalization with CMV MP-mCherry, PDLP1-GFP (*Thomas et al., 2008*) or YFP-PDCB1 (*Simpson et al., 2009*). To simultaneously visualize AtFH2 and actin filaments, GFP-ABD2 marker (*Sheahan et al., 2004*) was introduced into *Arabidopsis* plants expressing *pCAMBIA1301-35S:AtFH2-mCherry* by genetic crossing. To determine whether the

organization of actin filaments is required for the localization of AtFH2, 7-day-old *Arabidopsis* seedlings were treated with 1 µM latrunculin B for 1 hr before imaging.

## Aniline blue staining and quantification

Plasmodesmata was revealed by staining with aniline blue. To observe colocalization of AtFH2 and callose, 7-day-old *Arabidopsis* seedlings were used. To quantify the accumulation of callose at PD, the fifth leaves of 4-week-old *Arabidopsis* were used. Briefly, *Arabidopsis* seedlings or detached leaves were initially treated with a solution containing 300 µM MBS for 30–60 min, which was followed by incubation with aniline blue solution (0.1% aniline blue in double-distilled water and 1 M glycerol, pH 9.5, at a volume ratio of 2:3) for 30 min at room temperature as described previously (*Levy et al., 2007*; *Simpson et al., 2009*). To quantify the accumulation of callose at PD in WT, *atfh1*, *atfh2* and *atfh1 atfh2* plants, confocal images were subjected to intensity analysis using ImageJ software (http://rsbweb.nih.gov/ij/; version 1.51) as described previously (*Simpson et al., 2009*). The average fluorescence intensity and the size of the aniline blue-stained callose foci were measured and compared between WT and formin loss-of-function mutants. For each genotype, more than 100 data sets were collected from at least 20 separate images. As the data did not show a normal distribution, statistical comparison between different genotype was performed with a Mann-Whitney U test using IBM SPSS Statistics version 25 software.

## Visualization and quantification of the organization of actin filaments in *Arabidopsis* leaf epidermal cells

To capture the organization of actin filaments in leaf epidermal cells, 7-day-old *Arabidopsis* seedlings expressing *35S:GFP-ABD2* were observed under a laser scanning confocal microscope and the Z series images were collected with the step size set at 0.5 µm. The density of actin filaments in the leaf epidermal cells was determined by measuring the value of occupancy as described previously (*Higaki et al., 2010*). Briefly, the Z-stack images were converted to binary images by thresholding and were subsequently skeletonized and processed with maximum intensity projection. The occupancy is determined as the proportion of the pixels representing filaments out of the total pixels in the selected region. More than 50 data sets were collected from at least 10 plants for each genotype. All measurements and image processing were performed by ImageJ software (version 1.51 s, http://imagej.nih.gov/ij).

## Observation of cytoplasmic streaming

Cytoplasmic streaming was quantified according to the method described previously (*Okamoto et al., 2016*). The hypocotyl cells were observed in a bright field under an IX71 microscope (Olympus) equipped with a × 40 objective. Digital images were collected at 1 s intervals for 10–15 min with a Retiga Exi Fast 1394 CCD camera (QImaging) using Image-Pro Express 6.3 software. For estimation of the velocity, more than 50 data sets were collected from at least five plants for each genotype. The moving plastids were randomly selected and measured using ImageJ software.

## Assays for diffusion of GFP and cell-to-cell movement of CMV MP-GFP

Cell-to-cell trafficking was examined by determining the diffusion of eGFP across cells or the cell-to-cell movement of CMV MP-eGFP, which is described in more detail at Bio-protocol (*Diao et al., 2019*). In brief, the transient expression of eGFP or CMV MP-eGFP was achieved by a biolistic DNA delivery system in *Arabidopsis* leaves. Bombardment of leaves with plasmid DNA was performed basically as described by *Liarzi and Epel (2005)* with slight modifications. Briefly, approximately 2 µg of plasmid DNA were mixed with 0.5 mg gold particles (diameter, 1 µm, Bio-Rad) in the presence of 20 µL of fresh N-[3-aminopropyl]−1,4-butanediamine (spermidine) (0.1 M) and 50 µL of $CaCl_2$ (2.5 M) before vortexing uninterruptedly for 3 min. The mixture was centrifuged for 10 s before precipitating for 5 min. The pellet was subsequently washed with 70% ethanol and absolute ethanol, and finally resuspended in 15 µL absolute ethanol. Rosette leaves from 3-week-old *Arabidopsis* plants which had not yet bolted were placed in a petri dish containing 20 mL of MS medium with 0.8% agar before bombardment. The target leaf was placed at a distance of 12 cm from the gene gun. The leaves were incubated for 24 hr or 48 hr at 23°C in the dark. To quantify the number of layers of

cells into which eGFP diffused, images of eGFP in the lower epidermis cells were captured by a Zeiss LSM 510 META at 24 hr and 48 hr. Statistical analysis was performed as described by *Levy et al. (2007)*. The cell that was transformed by DNA bombardment was defined as layer 0, and the cells that share a common cell wall with layer 0 were defined as layer 1. Cells that share a common cell wall with layer 1 cells, but not with layer 0 cells, were defined as layer 2 cells. Cells that expressed GFP but showed no diffusion were not counted to avoid the situation in which damage was caused by the bombardment. To indicate the bombarded cell, *pdGN-35S:HDEL-mCherry* plasmid was used for bombardment along with the *pdGN* plasmid expressing eGFP. Both *pdGN* and *pdGN-35S: HDEL-mCherry* plasmids at 2 µg were mixed with 1 µm gold particles (Bio-Rad) for the bombardment. Results were analyzed with the nonparametric Mann-Whitney U test using IBM SPSS Statistics version 25 software.

## Fny-CMV inoculation and quantification of infection in *Arabidopsis* leaves

Fny-CMV infected *N. benthamiana* leaves were used for virus purification. After 7 days, the extract of virions at a concentration of 100 µg/ml was rub inoculated onto *Arabidopsis* seedlings at the 5–6 true-leaf stage (*Du et al., 2014*). The extent of infection was assessed by determining the amount of CMV *MP* RNA and quantifying the morphology of leaves. To detect the amount of CMV *MP* RNA in plants, a quantitative RT-PCR was performed. Primers Oligo $(dT)_{18}$ and CMV $MP_{REV}$ were used to reverse transcribe plant total RNA and RNA of CMV *MP* respectively. Primer pairs CMV *MP*-$RT_{FOR}$/ CMV *MP*-$RT_{REV}$ (*Supplementary File 1*) were used to determine the transcript levels of CMV *MP*. The relative amount of CMV *MP* transcripts was quantified using the $2^{-\Delta\Delta Ct}$ method with *eIF4A* as the internal control.

## Protein purification

AtFH2-FH2, AtFH2-FH1FH2, AtFH2ΔN and AtFH2-FH2M either fused with GST or 6 × His were expressed in the *E. coli* Tuner (DE3) pLysS strain by induction with 0.4 mM isopropyl β-D-thiogalacto-pyranoside overnight at 16°C. *E. coli* cells were collected by centrifugation at 6000 rpm (Beckman JA-10), and resuspended either in 1 × PBS (140 mM NaCl, 2.7 mM KCl, 10 mM $Na_2HPO4$, 1.8 mM $KH_2PO4$) or in 1 × binding buffer (25 mM Tris-HCl, pH 8.0, 250 mM KCl, 5 mM imidazole, 2 mM DTT) with proteinase inhibitor cocktail for the purification of GST fusion proteins or His-tag fusion proteins. After sonication and centrifugation at 27,216 g for 30 min, the supernatant was incubated with glutathione-sepharose (Amersham) or Ni-NTA resin (Novagen), using purification procedures according to the manufacturers' instructions. All purified proteins were dialyzed against 5 mM Tris-HCl, pH 8.0, 5% glycerol, flash frozen in liquid nitrogen, and stored at −80°C. They were preclarified at 200,000 g for 30 min at 4°C before the subsequent analyses. Actin was isolated from acetone powder of rabbit skeletal muscle (*Spudich and Watt, 1971*). Monomeric Ca-ATP-actin was subsequently purified by chromatography on Sephacryl S-300 at 4°C in Buffer G (5 mM Tris-HCl, pH 8.0, 0.2 mM ATP, 0.1 mM $CaCl_2$, 0.5 mM DTT, 0.1 mM $NaN_3$) (*Pollard, 1984*). Actin was labeled on Cys-374 with pyrene iodoacetamide to monitor the kinetic process of actin polymerization and depolymerization (*Pollard, 1984*) or labeled with Oregon-green as described previously (*Amann and Pollard, 2001*). Human profilin I was purified as described previously (*Fedorov et al., 1994*), and AtFH1-FH1FH2 was purified according to *Michelot et al. (2005)*.

## Actin nucleation assay

The actin nucleation assay procedure was performed as described previously (*Higgs et al., 1999*). Mg-ATP-actin or actin-profilin complexes (3 µM, 10% pyrene labeled) were incubated with various concentrations of AtFH2 for 3 min at room temperature before the addition of one-tenth volume of 10 × KMEI (500 mM KCl, 10 mM $MgCl_2$, 10 mM EGTA, 100 mM imidazole-HCl, pH 7.0). Polymerization of actin was traced by monitoring pyrene fluorescence by a QuantaMaster Luminescence QM 3 PH fluorometer (Photo Technology International, Inc.) with the excitation and emission wavelengths set at 365 nm and 407 nm, respectively.

## Seeded actin elongation assay

A seeded actin elongation assay, to determine the affinity of AtFH2 for the barbed end of actin filaments, was performed roughly according to the method described previously (*Huang et al., 2003*). Actin filaments at a final concentration of 0.8 μM were incubated for 3 min at room temperature with various concentrations of AtFH2. Actin elongation was initiated by the addition of actin-profilin complexes (1 μM, 10% pyrene labeled) to the actin filament mixture. The equilibrium binding constant ($K_d$) of AtFH2 proteins for the barbed end of actin filaments was determined by plotting the initial elongation rate as a function of the concentration of AtFH2 proteins using the following equation:

$$V_i = V_{if} + (V_{ib} - V_{if}) \left( \frac{K_d + [\text{ends}] + [\text{AtFH2}] - \sqrt{(K_d + [\text{ends}] + [\text{AtFH2}])^2 - 4[\text{ends}][\text{AtFH2}]}}{2[\text{ends}]} \right)$$

where $V_i$ is the observed rate of elongation, $V_{if}$ is the rate of elongation when all the barbed ends are free, $V_{ib}$ is the rate of elongation when all the barbed ends are capped, [ends] is the concentration of barbed ends, and [AtFH2] is the concentration of AtFH2. The data were modeled with KaleidaGraph software (version 4.03, http://kaleidagraph.software.informer.com/).

## Actin depolymerization assay

The dilution-mediated actin depolymerization assay was performed as described previously (*Huang et al., 2003*). 5 μM F-actin (50% pyrene-labeled) was incubated with various concentrations of AtFH2 for 3 min at room temperature before being diluted 25-fold into Buffer G. Actin depolymerization was traced by monitoring the changes in pyrene fluorescence by a QuantaMaster Luminescence QM 3 PH fluorometer (Photo Technology International, Inc.) with the excitation and emission wavelengths set at 365 nm and 407 nm, respectively.

## Fluorescence light microscopy of actin filaments

The annealing of actin filaments was visualized and quantified by fluorescence light microscopy as described previously (*Huang et al., 2003*). Briefly, actin filaments were labeled with equimolar rhodamine-phalloidin (Sigma-Aldrich) and then they were broken by sonication in the absence or presence of different concentrations of AtFH2. After incubation for a certain period of time, actin filaments were visualized under an Olympus IX71 microscope equipped with a × 60, 1.42–numerical aperture oil objective and the images were acquired by a Retiga EXi Fast 1394 CCD camera (QImaging) with Image-Pro Express 6.3 software. The effect of AtFH2 on actin filament annealing was quantified by measuring the length of actin filaments.

## Direct visualization of actin filament dynamics by total internal reflection fluorescence microscopy (TIRFM)

The effect of AtFH2 on the dynamics of single actin filaments was determined by TIRFM as described previously (*Zheng et al., 2012*). The effect of AtFH2 on actin nucleation was analyzed by counting the number of actin filaments within microscope fields, and the effect of AtFH2 on the growth of actin filaments was analyzed by performing kymograph analysis by tracing the elongating end of actin filaments using ImageJ software (http://rsbweb.nih.gov/ij/; version 1.51).

## Size exclusion chromatography

Size exclusion chromatography was performed to determine the size of AtFH2-FH2 and AtFH2-FH2M. Protein samples in 1 mL of running buffer (100 mM Tris-HCl, pH 8.5, 150 mM NaCl, 1 mM DTT and 5% glycerol) were loaded onto a Superdex 200 Increase 10/300 gel filtration column (GE) pre-equilibrated with running buffer, and the column was run at the speed of 0.4 mL/min with running buffer. Protein fractions of 0.5 ml were collected and separated by SDS-PAGE. Molecular weights were determined according to the manufacturer's instructions.

## Statistical analyses

The statistical analysis of the datasets was performed using IBM SPSS Statistics version 25 software. The normality of the datasets was initially assessed by Shapiro-Wilk tests. If the data were normally

distributed, the Student's t-test was applied for the subsequent statistical analyses. However, if the data were not normally distributed, the Mann-Whitney U test was applied for the subsequent statistical analyses. Differences were considered significant when $p<0.05$ and differences were considered extremely significant when $p<0.01$. The statistical tests and number of replicates are provided in the figure legend.

## Acknowledgements

We thank Andy Maule (John Innes Center, UK) for the marker lines expressing *PDLP1pro:PDLP1-GFP* and *PDCB1pro:YFP-PDCB1*, and Laurent Blanchoin and Alphe´e Michelot (Biosciences and Biotechnology Institute of Grenoble, France) for the AtFH1-FH1FH2 prokaryotic expression plasmid and great support on this project. We also thank NASC for providing sequence-indexed T-DNA insertion lines. This work was supported by grants from National Natural Science Foundation of China (31671390, 31121065 and 31471266). The work in Huang's lab is also partially supported by startup funding from Tsinghua-Peking Joint Center for Life Sciences.

## Additional information

### Funding

| Funder | Grant reference number | Author |
|---|---|---|
| National Natural Science Foundation of China | 31471266; 31671390 | Shanjin Huang |
| National Natural Science Foundation of China | 31121065 | Yule Liu |

The funders had no role in study design, data collection and interpretation, or the decision to submit the work for publication.

### Author contributions

Min Diao, Data curation, Software, Formal analysis, Validation, Investigation, Visualization, Methodology, Writing—original draft; Sulin Ren, Software, Formal analysis, Validation, Investigation, Methodology; Qiannan Wang, Software, Formal analysis, Investigation, Visualization, Methodology; Lichao Qian, Resources, Investigation, Methodology; Jiangfeng Shen, Software, Formal analysis, Validation, Investigation; Yule Liu, Resources, Data curation, Formal analysis, Supervision; Shanjin Huang, Conceptualization, Resources, Data curation, Supervision, Funding acquisition, Writing—original draft, Writing—review and editing

### Author ORCIDs

Min Diao  http://orcid.org/0000-0002-7960-9710
Yule Liu  http://orcid.org/0000-0002-4423-6045
Shanjin Huang  http://orcid.org/0000-0001-9517-2515

### Decision letter and Author response

Decision letter https://doi.org/10.7554/eLife.36316.060
Author response https://doi.org/10.7554/eLife.36316.061

## Additional files

### Supplementary files

• Supplementary file 1. Primer form
DOI: https://doi.org/10.7554/eLife.36316.057

• Transparent reporting form
DOI: https://doi.org/10.7554/eLife.36316.058

## Data availability

All data generated or analysed during this study are included in the manuscript and supporting files. Source data files have been provided for related Figures shown in the manuscript.

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
