## [Decision Letter]

Thank you for submitting your article "*Arabidopsis* Formin 2 Regulates Cell-to-Cell Trafficking by Capping and Stabilizing Actin Filaments at Plasmodesmata" for consideration by *eLife*. Your article has been reviewed Christian Hardtke as the Senior Editor, a Reviewing Editor, and three reviewers. The reviewers have opted to remain anonymous.

The reviewers have discussed the reviews with one another and the Reviewing Editor has drafted this decision to help you prepare a revised submission.

This manuscript presents the finding that several *Arabidopsis* Class I formins localise to plasmodesmata. Focusing on AtFH2 (targeted to plasmodesmata by its transmembrane domain) the authors find that it does not nucleate actin filaments but rather caps and stabilises actin filaments. AtFH1 is a known nucleator of actin filaments but also localises to plasmodesmata. Both AtFH2 and AtFH1 play a role in the regulation of the movement of GFP and are more susceptible to CMV infection, and mutants in both formins show greater movement of CMV MP between cells. The authors ultimately propose a model in which F-actin is present in the cytoplasmic sleeve of plasmodesmata, anchored by AtFH2. They propose that loss of the AtFH2 anchor increases the movement of GFP between cells by simply removing a physical blockage caused by F-actin. This manuscript presents a new actin-associated protein that associates at plasmodesmata, adding to actin, myosin, NET1A and Arp3. The authors have elegantly shown that AtFH1 and AtFH2 have nucleation/assembly and capping roles respectively, suggesting that there is specific regulation of F-actin at plasmodesmata.

Summary:

The authors present convincing data that the At formin proteins localize to the plasmodesmata and play a role in controlling size exclusion. The work appears to be well done but the results are over interpreted, especially with regard to the models that they draw from their data.

Essential revisions:

The manuscript was reviewed by three knowledgeable reviewers. The consensus was that the paper would be suitable for publication if some revisions were made and a few additional experiments completed. Perhaps the strongest, negative comments related to the model proposed with regard to AtFH2. The feeling was that this model was not well supported by the data presented. Here are some of the specific comments made by the reviewers that relate to this:

The model in Figure 7J suggests there are multiple F-actin filaments in the cytoplasmic sleeve of a plasmodesma. Given that F-actin has a diameter of ~6nm, and a cytoplasmic sleeve might be only ~10-20nm at its widest point (doi:10.1038/nplants.2017.82), this seems an unlikely model. The authors say they can't directly visualise actin in plasmodesmata, but new advances in high resolution live imaging, or electron microscopy, would allow the domain of localisation of AtFH2 (within the plasmodesma or at the neck) to be assessed. It would further allow the abundance of actin in plasmodesmata to be assessed in the *atfh2* mutant. The model needs better justification, or other models should be considered.

There is no investigation or discussion of what the role of AtFH2 and actin in plasmodesmata is. Simply blocking up the channel doesn't relate to structure or function of plasmodesmata in any specific way.

Hence, we suggest removing the model completely from the paper or, at least, toning it down considerably by offering alternative explanations that also fit the dataset.

Additional major concerns with the paper are as follows:

1) PD callose is also one of the key factors involved in modulating PD SEL. Have authors checked the PD callose level in different genotypes, single and double mutants along with wild type? It is known that GFP movement from cell to cell can be regulated by callose deposition, does AtFH2 affect this? It's also known that actin regulates cyoplasmic streaming which would influence rate of GFP 'diffusion' between cells, does AtFH2 affect cytoplasmic streaming?

2) The experiment in which the *atfh2* mutant is complemented by plasmodesmata-targeted FIMBRIM 5 seems confounded. FIM5 and AtFH2 are functionally different – while a similar phenotype may be observed no conclusions can be drawn regarding the mode of action of AtFH2 at plasmodesmata. Show localisation of PDFIM5.

3) Provide the control showing the arrangement of actin filaments when AtFH2 is ectopically expressed. Please provide information whether AtFH2 over-expression lines showing any PD trafficking-related phenotype?

4) Authors have used the truncated domain for checking the biochemical properties of AtFH2, I wonder whether these domains individually complement the *atfh2* phenotype in planta. How we know whether this AtFH2△N function in the same way with membrane attached full-length FH2? A related question…is the TM domain sufficient to target proteins to the PD and, if yes, is it also sufficient for the phenotypes seen?

5) The bombardment experiments should have been executed with a non-mobile fluorescent protein that marks the bombarded cell to ensure that movement is correctly interpreted. If the authors have not done this, we recognise that this would require the data set to be entirely repeated. Therefore, we suggest the following alternative experiment – that the authors provide a control dataset in which the bombardments are performed with and without a transformation marker (e.g. ER-RFP) to provide supporting evidence that misinterpretation of movement is not frequent under their experimental conditions.

[Editors' note: further revisions were requested prior to acceptance, as described below.]

Thank you for resubmitting your work entitled "*Arabidopsis* Formin 2 Regulates Cell-to-Cell Trafficking by Capping and Stabilizing Actin Filaments at Plasmodesmata" for further consideration at *eLife*. Your revised article has been favorably evaluated by Christian Hardtke (Senior Editor), a Reviewing Editor, and two reviewers.

The manuscript has been improved but there are some remaining issues that need to be addressed before acceptance, as outlined below:

*Reviewer #2*:

I am happy to see the manuscript revised appropriately according to editor's and reviewers' comments. I have no any major comments.

*Reviewer #3*:

The revised manuscript from Diao et al., has addressed many of my comments and I think it reads well. I am pleased to see they have removed the model. However, some of the points included in the list of concerns have not been addressed and I hope they are before the paper is accepted.

- I am not convinced by the authors’ argument for including the PDFIM5 experiment. I still consider the experiment is confounded because: (1) the authors have not experimentally shown that PDFIM5 stabilises actin filaments; (2) FIM and AtFH2 function in very different ways and as they haven't demonstrated that both proteins influence the actin at plasmodesmata in the same way they can't draw the conclusion that both proteins influence plasmodesmata by the same process. In my opinion, removing this experiment from the paper does not weaken it.

- The authors have not addressed the comment that AtFH1 and AtFH2 are not redundant; it is still mentioned in the Abstract that they are and I consider this incorrect interpretation of the data they present. Given that the *atfh1* and *atfh2* single mutants both exhibit phenotypes, and that AtFH1 and AtFH2 have different biochemical functions, the authors can only conclude that they are partially redundant (if at all). With respect to the claim that AtFH1 has capping activity the authors refer to Michelot et al., 2005. This paper makes the statement "The FH2-Cter domain acts like an efficient capper and blocks elongation at the barbed end of actin filaments. However, the presence of the FH1 domain allows the FH1-FH2-Cter protein to bind the barbed ends and allow elongation at the barbed ends." This implies to me that the full length AtFH1 does not have good capping activity, and therefore that AtFH1 and AtFH2 are unlikely to be fully redundant.

- The authors are also still referring to CMV MP-GFP movement from cell-to-cell as diffusion. CMV MP-GFP clearly localises at plasmodesmata and is therefore not freely mobile or 'diffusing' in the cell. It is well accepted that viral MPs are active and specific in their translocation from cell to cell via plasmodesmata. The use of the term diffusion here is incorrect and I recommend changing it to cell-to-cell movement CMV MP-GFP.

- The authors have changed their statistical tests for data they state is not normal. However, details of the test they used to identify that the data are not normal have not been supplied.

- The source files are simply tables that contain the results from the analysis of the data, not the raw data. They offer nothing new that isn't in the manuscript other than a p-value.

---

## [Author Response]

The manuscript was reviewed by three knowledgeable reviewers. The consensus was that the paper would be suitable for publication if some revisions were made and a few additional experiments completed. Perhaps the strongest, negative comments related to the model proposed with regard to AtFH2. The feeling was that this model was not well supported by the data presented. Here are some of the specific comments made by the reviewers that relate to this:The model in Figure 7J suggests there are multiple F-actin filaments in the cytoplasmic sleeve of a plasmodesma. Given that F-actin has a diameter of ~6nm, and a cytoplasmic sleeve might be only ~10-20nm at its widest point (doi:10.1038/nplants.2017.82), this seems an unlikely model. The authors say they can't directly visualise actin in plasmodesmata, but new advances in high resolution live imaging, or electron microscopy, would allow the domain of localisation of AtFH2 (within the plasmodesma or at the neck) to be assessed. It would further allow the abundance of actin in plasmodesmata to be assessed in the atfh2 mutant. The model needs better justification, or other models should be considered.

We used super resolution Structured Illumination Microscopy (SIM) to determine the localization of AtFH2, but we failed to obtain useful additional information compared to that previously captured by laser scanning confocal microscopy. This is very likely because the size of PD (with a length of about 150 nm across cells and a width of roughly about 20-30 nm) is too small to be imaged by SIM (which has a resolution of about 85 nm). Therefore, SIM cannot yield any useful information in terms of the precise localization of AtFH2. We also attempted to determine the localization of AtFH2 with immunogold electron microscopy using anti-GFP antibody, but it did not work well in our hands.

In addition, we found that it is almost impossible to specifically visualize actin filaments at PD with SIM because actin filaments are very dense at the border of cells (see Figure 1—figure supplement 2C; Figure 2—figure supplement 3C). Therefore, we do not have direct evidence to judge whether the density of actin filaments is reduced at PD in *atfh2* mutants. We agree that, with the current set of data, the model is not fully supported, and we therefore deleted the model in this revised manuscript. Accordingly, we rewrote the statement about the potential function of AtFH2 in regulating actin dynamics at PD in the text (Introduction; Discussion section.

There is no investigation or discussion of what the role of AtFH2 and actin in plasmodesmata is. Simply blocking up the channel doesn't relate to structure or function of plasmodesmata in any specific way.

Our data suggest that the amount of actin filaments and/or their stability is crucial for the permeability of PD. Besides speculating that actin filaments might function as the barrier in blocking cell-to-cell trafficking as proposed previously, we really do not know how AtFH2 and actin might play any specific role in regulating the permeability of PD. Given that the reconstructed ultrastructure of PD using electron tomography showed that the contact between the PM and ER is highly dynamic and might be crucial for the permeability of PD (doi:10.1038/nplants.2017.82), we speculate that AtFH2-tethered actin filaments might be involved in regulating the contact between PM and ER. The detailed information can be found in the Discussion section.

Hence, we suggest removing the model completely from the paper or, at least, toning it down considerably by offering alternative explanations that also fit the dataset.

We totally agree with the comments that it is inappropriate to fit more actin filaments (with their diameter of about 7 nm) into the cytoplasmic sleeves of PD (with its diameter of less than 10 nm) (doi:10.1038/nplants.2017.82). Considering this along with the fact that we do not know exactly where AtFH2 is localized in PD, we decide to remove the model. Accordingly, we modified our statement about the potential function of AtFH2 and actin at PD. In this revised manuscript, we simply argue that AtFH2-mediated stability of actin filaments and/or the amount of actin filaments at PD might be crucial for the regulation of SEL of PD.

1) PD callose is also one of the key factors involved in modulating PD SEL. Have authors checked the PD callose level in different genotypes, single and double mutants along with wild type? It is known that GFP movement from cell to cell can be regulated by callose deposition, does AtFH2 affect this? It's also known that actin regulates cyoplasmic streaming which would influence rate of GFP 'diffusion' between cells, does AtFH2 affect cytoplasmic streaming?

We performed callose staining for WT and *atfh2, atfh1* and *atfh1fh2* mutants, and found that there is no dramatic difference in the accumulation of callose at PD between formin loss-of-function mutants and WT (see the results in the newly prepared Figure 4—figure supplement 2). In addition, we determined the velocity of cytoplasmic streaming in *Arabidopsis* hypocotyl cells and found that it is similar between WT and *atfh2* mutants, which suggests that the increase in SEL of PD does not result from the changes in intracellular trafficking (see the results in the newly prepared Figure 2figure supplement 2).

2) The experiment in which the atfh2 mutant is complemented by plasmodesmata-targeted Fimbrin5 seems confounded. FIM5 and AtFH2 are functionally different – while a similar phenotype may be observed no conclusions can be drawn regarding the mode of action of AtFH2 at plasmodesmata. Show localisation of PDFIM5.

Based on our in vitro biochemical results that AtFH2 caps and stabilizes actin filaments, we speculated that the stability and/or the amount of actin filaments at PD are reduced in *atfh2* mutants. In this regard, we hypothesized that if we could prevent the depolymerization of actin filaments in PD, we might be able to suppress the cell-to-cell trafficking phenotype in *atfh2*. We tested this hypothesis by introducing a PD-targeting actin filament stabilizer. We fused the PD-targeting N-terminus of AtFH2 with an actin filament stabilizer FIM5 and found that the fusion protein called PD-FIM5 indeed targets to PD (Figure 7I). We found that introduction of PDFIM5 into *atfh2* suppressed the cell-to-cell trafficking phenotype, which suggests that the cell-to-cell trafficking phenotype in *atfh2* mutants indeed results to some extent from the instability of actin filaments at PD.

3) Provide the control showing the arrangement of actin filaments when AtFH2 is ectopically expressed. Please provide information whether AtFH2 over-expression lines showing any PD trafficking-related phenotype?

We found that overexpression of *AtFH2* does not have an overt effect on the organization of actin filaments and cell-to-cell trafficking (see the results in the newly prepared Figure 2—figure supplement 3).

4) Authors have used the truncated domain for checking the biochemical properties of AtFH2, I wonder whether these domains individually complement the atfh2 phenotype in planta. How we know whether this AtFH2△N function in the same way with membrane attached full-length FH2? A related question…is the TM domain sufficient to target proteins to the PD and, if yes, is it also sufficient for the phenotypes seen?

The actin-related function of formin proteins is determined by the FH1FH2 domain. Therefore, colleagues in this field have routinely analyzed the biochemical activity of the formin FH1FH2 domain in regulating actin dynamics in vitro in order to understand the biochemical activity of formin proteins in regulating actin dynamics. In this regard, the FH1FH2 domain can to some extent represent the full-length protein in terms of the activity in regulating actin dynamics.

We demonstrate that the TM domain of AtFH2 is sufficient and necessary for the targeting of AtFH2 to PD (Figure 1). In further support of this notion, we found that the TM domain-containing N-terminus of AtFH2 is able to direct the heterologous *Arabidopsis* FIM5 to PD (Figure 7I). However, we found that simply targeting AtFH2 to PD is not sufficient for its function, since AtFH2M is able to target to PD but fails to rescue the cell-to-cell trafficking phenotype in *atfh2*.

5) The bombardment experiments should have been executed with a non-mobile fluorescent protein that marks the bombarded cell to ensure that movement is correctly interpreted. If the authors have not done this, we recognise that this would require the data set to be entirely repeated. Therefore, we suggest the following alternative experiment – that the authors provide a control dataset in which the bombardments are performed with and without a transformation marker (e.g. ER-RFP) to provide supporting evidence that misinterpretation of movement is not frequent under their experimental conditions.

As suggested, we performed the bombardments using the eGFP-expressing plasmid *pdGN* along with a plasmid expressing the non-mobile HDEL-mCherry (*pdGN-35S:HDEL-mCherry*). We found that the HDEL-mCherry signal mostly stays in one cell while the eGFP signal appears in other cells besides the cell containing the HDEL-mCherry signal. We conclude that the cell having the non-mobile HDEL-mCherry signal is the bombarded cell. Therefore, the eGFP signal in the surrounding cells results from diffusion. We repeated the eGFP diffusion experiments using HDEL-mCherry to indicate the bombarded cell and compared the PD permeability between WT and the formin loss-of-function mutants and obtained the same results as before (see the results in the newly prepared Figure 2—figure supplement 1).

[Editors' note: further revisions were requested prior to acceptance, as described below.]

The manuscript has been improved but there are some remaining issues that need to be addressed before acceptance, as outlined below:Reviewer #3:[…] - I am not convinced by the authors’ argument for including the PDFIM5 experiment. I still consider the experiment is confounded because: (1) the authors have not experimentally shown that PDFIM5 stabilises actin filaments; (2) FIM and AtFH2 function in very different ways and as they haven't demonstrated that both proteins influence the actin at plasmodesmata in the same way they can't draw the conclusion that both proteins influence plasmodesmata by the same process. In my opinion, removing this experiment from the paper does not weaken it.

FIM5 was previously demonstrated to stabilize actin filaments in vitro (Wu et al., 2010), which allows us to speculate that PD-FIM5 will stabilize actin filaments at PD since it is indeed directed to PD by its N-terminal PD targeting sequence. Although AtFH2 and PD-FIM5 act through different mechanisms, we could speculate that both proteins will stabilize actin filaments and influence the amount of actin filaments at PD. However, we cannot currently demonstrate that both proteins influence PD by the same process. Therefore, for clarity, we removed the data as suggested. We also removed the related information in the Results section, legends, Materials and methods section and primer list sections.

- The authors have not addressed the comment that AtFH1 and AtFH2 are not redundant; it is still mentioned in the Abstract that they are and I consider this incorrect interpretation of the data they present. Given that the atfh1 and atfh2 single mutants both exhibit phenotypes, and that AtFH1 and AtFH2 have different biochemical functions, the authors can only conclude that they are partially redundant (if at all). With respect to the claim that AtFH1 has capping activity the authors refer to Michelot et al., 2005. This paper makes the statement "The FH2-Cter domain acts like an efficient capper and blocks elongation at the barbed end of actin filaments. However, the presence of the FH1 domain allows the FH1-FH2-Cter protein to bind the barbed ends and allow elongation at the barbed ends." This implies to me that the full length AtFH1 does not have good capping activity, and therefore that AtFH1 and AtFH2 are unlikely to be fully redundant.

We agree with this reviewer that AtFH1 and AtFH2 are only partially redundant in terms of their biochemical activities in regulating actin dynamics. We therefore modified our statement accordingly.

- The authors are also still referring to CMV MP-GFP movement from cell-to-cell as diffusion. CMV MP-GFP clearly localises at plasmodesmata and is therefore not freely mobile or 'diffusing' in the cell. It is well accepted that viral MPs are active and specific in their translocation from cell to cell via plasmodesmata. The use of the term diffusion here is incorrect and I recommend changing it to cell-to-cell movement CMV MP-GFP.

We fixed the problem throughout the manuscript.

- The authors have changed their statistical tests for data they state is not normal. However, details of the test they used to identify that the data are not normal have not been supplied.

We initially assessed the datasets with Shapiro-Wilk tests to see whether they have a normal distribution. We have now included this test information in the source data and we also added a statement about how we performed the statistical analysis in the Materials and methods section.

- The source files are simply tables that contain the results from the analysis of the data, not the raw data. They offer nothing new that isn't in the manuscript other than a p-value.

Sorry, we overlooked this question during the last revision, and we have now included the raw data. We also explained that we used Shapiro-Wilks tests to assess the datasets to see whether they have a normal distribution. We have included this information in the newly prepared source data.